# Sharp Analysis for KL-Regularized Contextual Bandits and RLHF

## Abstract

*Reverse-Kullback-Leibler* regularization has emerged to be a predominant technique used to enhance policy optimization in reinforcement learning (RL) and reinforcement learning from human feedback (RLHF), which forces the learned policy to stay close to a reference policy. While the effectiveness and necessity of KL-regularization have been empirically demonstrated in various practical scenarios, current theoretical analysis of KL-regularized RLHF still obtains the same $\mathcal{O}(1/\epsilon^2)$ sample complexity as problems without KL-regularization. To understand the fundamental distinction between policy learning objectives with KL-regularization and ones without KL-regularization, we are the first to theoretically demonstrate the power of KL-regularization by providing a sharp analysis for KL-regularized contextual bandits and RLHF, revealing an $\mathcal{O}(1/\epsilon)$ sample complexity when $\epsilon$ is sufficiently small.

We further explore the role of data coverage in contextual bandits and RLHF. While the coverage assumption is commonly employed in offline RLHF to link the samples from the reference policy to the optimal policy, often at the cost of a multiplicative dependence on the coverage coefficient, its impact on the sample complexity of online RLHF remains unclear. Previous theoretical analyses of online RLHF typically require explicit exploration and additional structural assumptions on the reward function class. In contrast, we show that with sufficient coverage from the reference policy, a simple two-stage mixed sampling strategy can achieve a sample complexity with only an additive dependence on the coverage coefficient. Our results provide a comprehensive understanding of the roles of KL-regularization and data coverage in RLHF, shedding light on the design of more efficient RLHF algorithms.

## 1 Introduction

Recently, *Reinforcement Learning from Human Feedback* (RLHF) has emerged as a central tool for aligning large language models (LLMs) and diffusion models with human values and preferences (Christiano et al., 2017; Ziegler et al., 2019; Ouyang et al., 2022; Bai et al., 2022; Rafailov et al., 2024), exhibiting impressive capabilities in applications, such as Chatgpt (Achiam et al., 2023), Claude (Anthropic, 2023), Gemini (Team et al., 2023), and LLaMA-3 (Meta, 2024).

RLHF methods treat the language model as a policy that takes a prompt $x$ and produces a response $a$ conditioned on $x$, and they optimize the policy by aligning it with human feedback. There are mainly two kinds of feedback: absolute rating and preference comparison. For absolute rating, the collection typically involves human annotators to provide rating scores like 1 to 5 (Wang et al., 2024a;b) for responses or hard 0-1 scores for math reasoning tasks since the reasoning tasks often have gold standard answers (Cobbe et al., 2021; Hendrycks et al., 2021; Xiong et al., 2024b).

On the other hand, preference comparison is frequently applied in chat tasks when making comparisons is much easier for human labeler (Achiam et al., 2023). Although preference feedback is believed to be more intuitive for human users and easier to collect, it also poses more challenges for the RLHF algorithms to effectively leverage the feedback signals since the reward signals are not directly observed. In practice, the learning process typically involves (a) constructing a reward model based on the maximum likelihood estimation (MLE) of *Bradley-Terry* (BT) (Bradley & Terry, 1952b) model from the preference feedback; (b) applying RL algorithms like PPO (Schulman et al.,

2017b) to train the language model so that it maximizes the reward signals with KL regularization (Ouyang et al., 2022; Bai et al., 2022; Touvron et al., 2023). There is also a series of work on direct preference learning algorithms, including Slic (Zhao et al., 2023b), DPO (Rafailov et al., 2024), IPO (Azar et al., 2024), which directly optimize the preference signals to train the language model without constructing the reward model. However, despite their efficiency, these types of algorithms often suffer from overfitting, and DPO suffers from a drop of chosen probability (Song et al., 2024; Liu et al., 2024; Amini et al., 2024; Huang et al., 2024).

Since human value and preference are so complicated that they are unlikely to be encompassed by the considered preference model classes, the learned model is easy to be hacked and become biased. Practically, the policy may generate disproportionate bold words or emoji to please the learned reward (Zhang et al., 2024). Hence, the KL-regularization between the learned policy and a reference policy (the pre-trained model after supervised fine-tuning) plays a fundamental role in RLHF to avoid overfitting. There is a line of RLHF work that realizes the significance of KL-regularization and regards the problem as a reverse-KL regularized contextual bandit (Ziegler et al., 2019; Wu et al., 2021; Ouyang et al., 2022; Rafailov et al., 2024; Xiong et al., 2024a; Ye et al., 2024b). However, they adopt the techniques from bandit framework and neglect the characteristic of reverse-KL-regularization, thus obtaining almost the same sample complexity with problems without KL-regularization. Therefore, the question of *whether there exists a fundamental distinction between policy learning objectives with and without KL-regularization* is still largely under-explored.

Compared to the offline RLHF algorithms (Rafailov et al., 2024; Azar et al., 2024; Chen et al., 2024) that can only use planning to approximate the solution to the relative entropy minimization problem (Ziebart et al., 2008; Song et al., 2024), online RLHF has been demonstrated to outperform offline methods empirically and theoretically (Bai et al., 2023; Meta, 2024; Xiong et al., 2024a; Tajwar et al., 2024; Song et al., 2024; Wu et al., 2024), because it has further interactions with human or the preference oracle. Most standard theoretical online RL techniques apply optimism to balance exploration and exploitation (Abbasi-Yadkori et al., 2011; Wang et al., 2020; Jin et al., 2021). However, it is inefficient to implement exploration for practical RLHF algorithms. Meanwhile, an emerging line of offline RLHF literature highlights the coverage of the reference policy $\pi_0$. The coverage of $\pi_0$ refers to the ability of the model to generate diverse responses for a wide range of prompts. A model with good coverage can generalize well to unseen contexts and actions, which is essential for the learned reward function to also generalize well. In practice, this is evidenced by the fact that the simple best-of-$n$ sampling based on $\pi_0$ is competitive with the well-tuned PPO algorithm for general open-ended conversation tasks (Dong et al., 2023), and the fact that the $\pi_0$ can solve a majority of the math problems with multiple responses (Shao et al., 2024; Nakano et al., 2021). However, the theoretical understanding of the role of coverage in online RLHF is still largely understudied. Thus, it is natural to ask *if explicit exploration is necessary for online RLHF with good coverage of $\pi_0$* and *how the coverage of $\pi_0$ affects the sample complexity of online RLHF*.

In this paper, we answer the above questions by

- providing a novel fine-grained analysis for KL-regularized contextual bandits and RLHF, which adapts to the optimization landscape of the reverse-KL regularization and reveals a sharper sample complexity than the existing results,

- proposing an efficient 2-stage mixed sampling strategy for online RLHF with good coverage of $\pi_0$, which achieves sample complexity with only an additive dependence on the coverage coefficient.

## 1.1 OUR CONTRIBUTIONS

In this work, we make a first attempt to illustrate the statistical benefits of KL-regularization for policy optimization in contextual bandits and reinforcement learning (RL) from preference feedback. Our main contributions are summarized as follows:

- In Section 3, we formulate RLHF with absolute-rating feedback as a contextual bandit problem with KL-regularization. First, we provide a lower bound for the KL-regularized contextual bandit problem, which indicates that the sample complexity of the problem is $\Omega(\eta \log N_{\mathcal{R}}(\epsilon)/\epsilon)$ when $\epsilon$ is sufficiently small, where $N_{\mathcal{R}}(\epsilon)$ is the covering number of the reward function class and $\eta$ is the KL-regularization coefficient.

- Then we showcase a novel analysis to upper bound the suboptimality gap of the KL-regularized objective in contextual bandits, and propose a simple two-stage mixed sampling strategy for online

RLHF which achieves a sample complexity of $\mathcal{O}(\max(\eta^2 D^2, \eta/\epsilon) \log N_{\mathcal{R}}(\epsilon/\delta))$ when the reward scale is a constant, where $D$ is the coverage coefficient of the reference policy $\pi_0$ and $\delta$ is the confidence parameter. To the best of our knowledge, this is the first work to provide a sharp sample complexity for KL-regularized contextual bandits.

- In Section 4, we extend our analysis to reinforcement learning from preference feedback. We rigorously demonstrate that KL-regularization is essential for more efficient policy learning in RLHF with preference data. We further propose a two-stage mixed sampling strategy for online preference learning setting with good coverage of $\pi_0$, which achieves a sample complexity of $\mathcal{O}(\max(\eta^2 D^2, \eta/\epsilon) \log N_{\mathcal{R}}(\epsilon/\delta))$ when the reward scale is a constant.

## 1.2 PREVIOUS UNDERSTANDING OF KL-REGULARIZATION IN RL

While we mainly focus on the theoretical understanding of KL-regularization in RLHF, it is also worth mentioning that our analysis for KL-regularized contextual bandits also contributes to the theoretical understanding of the impact of KL-regularization in RL since contextual bandits can be viewed as a simplified version of Markov decision processes (MDPs).

In RL, KL-regularization has been widely used to stabilize the learning process and prevent the policy from deviating too far from the reference policy. Here, we provide a brief overview of the existing understanding of KL-regularization in decision-making problems. From the perspective of policy optimization, KL-regularization captures entropy regularization as a special case [1], which is also an extensively used technique in RL literature (Sutton, 2018; Szepesvári, 2022). There is a large body of literature that has explored the benefits of entropy regularization or KL-regularization in RL (Schulman et al., 2015; Fox et al., 2016; Schulman et al., 2017a; Haarnoja et al., 2017; 2018; Ahmed et al., 2019). Most related to our work, Ahmed et al. (2019) provided a comprehensive understanding of the role of entropy regularization in RL, showing that entropy regularization can improve the training efficiency and stability of the policy optimization process by changing the optimization landscape through experiments on continuous control tasks (Brockman, 2016).

Theoretically, Neu et al. (2017) provided a unified view of entropy regularization as approximate variants of Mirror Descent or Dual Averaging, and left the statistical justification for using entropy regularization in RL as an open question. Geist et al. (2019) provided a framework for analyzing the error propagation in regularized MDPs, which also focused on the proof of the convergence for the policy optimization methods with regularization and lacked a sharp sample complexity analysis.

## 1.3 OTHER RELATED WORK

**Analyses for Policy Optimization with Regularization** While it is previously unknown whether regularization can improve the sample complexity of policy optimization without additional assumptions, there are some works that provided a sharp convergence rate in the presence of regularization (Mei et al., 2020; Shani et al., 2020; Agarwal et al., 2020; 2021). However, these works either assumed the access of exact or unbiased policy gradient or required uniform value function approximation error, which are not the standard case in sample-based RL setting.

**RLHF Algorithms** There are mainly three types of RLHF algorithms: offline, online and hyrbid. The most well-known offline algorithms are Slic (Zhao et al., 2023b), Direct Preference Optimization (DPO) (Rafailov et al., 2024), Identity-PO (IPO) (Azar et al., 2024) and (SPIN) (Chen et al., 2024). They aim to approximate the closed-form solution of the optimization problem on a fixed offline dataset. For the online algorithms, the most representative one is Proximal Policy Optimization (PPO) (Schulman et al., 2017b). PPO has been used in the Chat-GPT (OpenAI, 2023), Gemini (Team et al., 2023), and Claude (Bai et al., 2022). However, the deep RL method PPO is known to be sample inefficient and unstable, making its success hard to reproduce for the open-source community. In response to this, there have been many efforts to propose alternative algorithms to the PPO algorithm. The Reward ranked fine-tuning (RAFT) (also known as rejection sampling finetuning) (Dong et al., 2023; Touvron et al., 2023; Gulcehre et al., 2023; Gui et al., 2024) is a stable framework requiring minimal hyper-parameter tuning, which iteratively learns from the best-of-n policy (Nakano et al., 2021). This framework proves to be particularly effective in the reasoning task such

---

[1]We can regard the entropy regularization as a special case of KL-regularization by setting the reference policy as the uniform distribution.

as (Gou et al., 2024; Tong et al., 2024). However, the RAFT-like algorithms only use the positive signal by imitating the best-of-n sampling. To further improve the efficiency, there is an emerging body of literature that proposes online direct preference optimization by extending DPO or IPO to an online iterative framework (Xiong et al., 2024a; Guo et al., 2024; Wu et al., 2024; Calandriello et al., 2024; Xiong et al., 2024b). Finally, for the third type, the common point of hybrid and online algorithms is that they both require further interaction with the preference oracle and on-policy data collection. The difference is that hybrid algorithms start with a pre-collected dataset (Xiong et al., 2024a; Song et al., 2024; Touvron et al., 2023), while the online algorithms learn from scratch.

**RLHF Theory**  The theoretical study of RLHF can date back to the dueling bandits (Yue et al., 2012) and follow-up work on MDPs (Wang et al., 2023a; Zhu et al., 2023). However, these works deviate from the practice because they do not realize the significance of KL-regularization and still choose the greedy policy that simply maximizes the reward. After this line of work, Xiong et al. (2024a); Ye et al. (2024b); Song et al. (2024) highlight the KL-regularization theoretically and incorporate the KL term into the learning objective. However, they circumvent the special advantages of KL-regularization and still follow the techniques in bandit analysis, thus obtaining loose bounds. In our paper, we establish a new lower bound and a sharper upper bound for the KL-regularized framework, thus validating the empirical advantage of KL-regularization. There are also some works extending KL-regularized RLHF from bandit problems to the Markov decision process (MDP) problems (Zhong et al., 2024; Xiong et al., 2024b). We expect that our techniques can also be extended to the MDP setting, which we leave for future work.

## 2 PRELIMINARIES

In this section, we formally state the problem settings of RL from human feedback (RLHF), where we consider two types of feedback: absolute rating and preference.

### 2.1 CONTEXTUAL BANDITS WITH KL REGULARIZATION

The first setting is the absolute-rating feedback, where we can query the ground-truth reward function to measure the quality of the responses by providing absolute reward value. For instance, in the NVIDIA Helpsteer project (Wang et al., 2023b; 2024c), human labelers are required to provide absolute score in five attributes: helpfulness, correctness, coherence, complexity, and verbosity. The dataset leads to many high-ranking open-source reward models, including the ArmoRM-Llama3-8B-v0.1 (Wang et al., 2024a;b), URM-LLaMa-3.1-8B[2], and Llama-3.1-Nemotron-70B-Reward[3]. We also notice that recently this feedback framework is extended to other task such as video generation (He et al., 2024).

The absolute-rating feedback is directly modeled as reward functions (Wang et al., 2024a; Xiong et al., 2024b), and is regarded as contextual bandits with KL regularization. In the contextual bandit setting, at each round $t$, the agent observes a context $x_t \in \mathcal{X}$ generated from a distribution $d_0$ and chooses an action $a_t \in \mathcal{A}$. The agent receives a stochastic reward $r_t \in \mathbb{R}$ depending on the context $x_t$ and the action $a_t$. The goal is to maximize the expected cumulative reward over $T$ rounds.

The learner has access to a family of reward functions $R(\theta, x, a)$ parameterized by $\theta \in \Theta$, such that there exists $\theta_* \in \Theta$ satisfying $\mathbb{E}[r_t|x_{1:t}, a_{1:t}] = R(\theta_*, x_t, a_t)$. WLOG, we assume that the reward feedback $r_t$ at all rounds is a non-negative real number bounded by $B$. We consider a KL-regularized objective as follows:

$$Q(\pi) = \mathbb{E}_{x \sim d_0} \mathbb{E}_{a \sim \pi(\cdot|x)} \left[ R(\theta_*, x, a) - \eta^{-1} \ln \frac{\pi(a|x)}{\pi_0(a|x)} \right], \quad (2.1)$$

where $\pi_0$ is a known fixed policy, and $\eta > 0$ is a hyperparameter that controls the trade-off between maximizing rewards and staying close to the reference policy $\pi_0$.

**Remark 2.1.** *It is worth noting that entropy or Kullback-Leibler (KL) regularization is also widely used in contextual bandits (Berthet & Perchet, 2017; Wu et al., 2016) and deep RL algorithms (Schulman et al., 2015; Fox et al., 2016; Schulman et al., 2017a; Haarnoja et al., 2017; 2018), where KL-divergence regularization is a popular technique for preventing drastic updates to the*

---

[2]https://huggingface.co/LxzGordon/URM-LLaMa-3.1-8B

[3]https://huggingface.co/nvidia/Llama-3.1-Nemotron-70B-Reward

*policy. Algorithms such as Trust Region Policy Optimization (TRPO) (Schulman et al., 2015) explicitly incorporate KL-regularization to limit the policy updates during optimization, ensuring that the updated policy does not deviate too much from the current policy. This constraint promotes stable and reliable learning, particularly in high-dimensional state-action spaces. Additionally, KL-regularization is central to Proximal Policy Optimization (PPO) (Schulman et al., 2017a), where a penalty term involving KL-divergence ensures updates remain within "trust region".*

## 2.2 Reinforcement Learning from Preference Feedback

The second framework we consider is preference feedback, which is widely applied in projects such as Chat-GPT (OpenAI, 2023) and Claude (Bai et al., 2022). Specifically, when receiving a prompt $x \in \mathcal{X}$, and two actions (responses) $a^1, a^2 \in \mathcal{A}$ from some LLM policy $\pi(\cdot|x)$, a preference oracle will give feedback $y$ defined as follows:

**Definition 2.2** (Preference Oracle). *A Preference Oracle is a function $\mathbb{P} : \mathcal{X} \times \mathcal{A} \times \mathcal{A} \to \{0, 1\}$. Given a context $x \in \mathcal{X}$ and two actions $a_1, a_2 \in \mathcal{A}$, the oracle can be queried to obtain a preference signal $y \sim Bernoulli(\mathbb{P}(x, a_1, a_2))$, where $y = 1$ indicates that $a_1$ is preferred to $a_2$ in the context $x$, and $y = 0$ indicates the opposite.*

To learn the preference, we follow Ouyang et al. (2022); Zhu et al. (2023); Rafailov et al. (2024); Liu et al. (2023); Xiong et al. (2024a) and assume that the preference oracle is measured by the difference of ground-truth reward functions $R(\theta_*, x, a)$, which is named the Bradley-Terry (BT) model (Bradley & Terry, 1952a).

**Definition 2.3** (Bradley-Terry Model). *Given a context $x \in \mathcal{X}$ and two actions $a_1, a_2 \in \mathcal{A}$, the probability of $a_1$ being preferred to $a_2$ is modeled as*

$$\mathbb{P}(x, a_1, a_2) = \frac{\exp(R(\theta_*, x, a_1))}{\exp(R(\theta_*, x, a_1)) + \exp(R(\theta_*, x, a_2))} = \sigma(R(\theta_*, x, a_1) - R(\theta_*, x, a_2)), \quad (2.2)$$

*where $\sigma(\cdot)$ is the sigmoid function.*

The RLHF training always follows the fine-tuning process, which yields a reference policy $\pi_0$. When performing RLHF on specific tasks, to avoid overfitting, we impose KL-regularization to the learned reward model when optimizing the policy. Hence, our objective function is also (2.1).

## 2.3 Additional Notations and Definitions

In this subsection, we introduce the definitions shared by both settings.

**Reward function class.** We consider a function class $\mathcal{R} = \{R(\theta, \cdot, \cdot) | \theta \in \Theta\}$ and for the realizability, we assume that the ground truth reward function $R(\theta_*, x, a)$ is in the function class $\mathcal{R}$. Then, we define the covering number of $\mathcal{R}$ as follows.

**Definition 2.4** ($\epsilon$-cover and covering number). *Given a function class $\mathcal{F}$, for each $\epsilon > 0$, an $\epsilon$-cover of $\mathcal{F}$ with respect to $||\cdot||_\infty$, denoted by $\mathcal{C}(\mathcal{F}, \epsilon)$, satisfies that for any $f \in \mathcal{F}$, we can find $f' \in \mathcal{C}(\mathcal{F}, \epsilon)$ such that $||f - f'||_\infty \leq \epsilon$. The $\epsilon$-covering number, denoted as $N_\mathcal{F}(\epsilon)$, is the smallest cardinality of such $\mathcal{C}(\mathcal{F}, \epsilon)$.*

**Planning oracle.** Given a reward model, we can learn the policy by optimizing the KL-regularized objective in (2.1). To simplify the analysis, we assume that there exists a planning oracle, which in empirical can be efficiently approximated by rejection sampling (Liu et al., 2023), Gibbs sampling (Xiong et al., 2024a), and iterative preference learning with a known reward (Dong et al., 2024).

**Definition 2.5** (Policy Improvement Oracle). *For a reward function $R(\theta, \cdot, \cdot) \in \mathcal{R}$ and a reference policy $\pi_0$, for any prompt $x \sim d_0$, we can compute:*

$$\pi_\theta^\eta(\cdot|x) := \underset{\pi(\cdot|x) \in \Delta(\mathcal{A})}{\arg\max} \mathbb{E}_{a \sim \pi(\cdot|x)} \left[ R(\theta, x, a) - \eta^{-1} \log \frac{\pi(a|x)}{\pi_0(a|x)} \right] \propto \pi_0(\cdot|x) \cdot \exp\left(\eta R(\theta, x, \cdot)\right).$$

Hence, the comparator policy is the solution of the oracle given the true reward function $R(\theta^*, \cdot, \cdot)$: $\pi^*(\cdot|\cdot) \propto \pi_0(\cdot|\cdot) \cdot \exp(\eta R(\theta^*, \cdot, \cdot))$. The **goal** is to minimize the sub-optimality of our learned policy $\widehat{\pi}$ with $\pi^*$: $Q(\pi^*) - Q(\widehat{\pi})$.

**Coverage conditions.** It is crucial to assume that our data-collector policy $\pi_0$ possesses good coverage, which can ensure that the learned reward function can generalize well to unseen contexts (prompts) and actions (responses), and thus can enable us to approximate the optimal policy. The global coverage is the uniform cover over all the policies in the considered class $\Pi$, which is standard in offline RL (Munos & Szepesvári, 2008; Song et al., 2024) and online RL (Xie et al., 2022; Rosset et al., 2024). Essentially, Song et al. (2024) demonstrated that global coverage is necessary for offline framework and Direct Preference Optimization (DPO) fails without global coverage. Hence, we introduce two types of global-coverage conditions.

**Definition 2.6** (Data Coverage). *Given a reference policy $\pi_0$, $D^2$ is the minimum positive real number satisfying $\forall\,(x, a) \in \mathcal{X} \times \mathcal{A}, s.t.\ \pi(a|x) > 0$ and $\forall\, b : \mathcal{X} \to [-B, B]$, we have*

$$\sup_{\theta, \theta' \in \Theta} \frac{|R(\theta', x, a) - R(\theta, x, a) - b(x)|^2}{\mathbb{E}_{x' \sim d_0} \mathbb{E}_{a' \sim \pi_0(\cdot|x')} |R(\theta', x', a') - R(\theta, x', a') - b(x')|^2} \leq D^2.$$

The coverage coefficient $D$ measures how well the in-sample error induced by distribution $d_0 \times \pi_0$ can cover the out-of-sample error, identifically speaking, it depicts the ability of $\pi_0$ to cover the action space. This concept is adapted from the F-design for online RL under general function approximation (Agarwal et al., 2024), and resembles the coverage coefficient for offline RL (Ye et al., 2024c;a), and the eluder dimension (Wang et al., 2020; Ye et al., 2023; Agarwal et al., 2023; Zhao et al., 2023a) for online RL.

**Definition 2.7** (Global-Policy Coverage). *Given a reference policy $\pi_0$, $C_{\mathrm{GL}}$ is the minimum positive real number satisfying that for any $\pi : \mathcal{X} \to \mathcal{A}$*

$$\sup_{x \sim d_0, a \in \mathcal{A}} \frac{\pi(a|x)}{\pi_0(a|x)} \leq C_{\mathrm{GL}}.$$

The two conditions require the reference policy to cover all possible policy distributions, which is standard and common in RL literature. Additionally, although the two conditions defined above are both global, it is obvious that $D^2 \leq C_{\mathrm{GL}}$, indicating that it is more general to assume a finite $D$.

Because of the KL-regularization for RLHF, the learned policy will not move too far from the reference policy. Hence, it is natural to relax the global coverage to local coverage inside the KL-ball (Song et al., 2024).

**Definition 2.8** (Local KL-ball Coverage). *Given a reference policy $\pi_0$, for a positive constant $\rho_{\mathrm{KL}} < \infty$, and all policy satisfying that $\mathbb{E}_{x \sim d_0}[\mathrm{KL}(\pi, \pi_0)] \leq \rho_{\mathrm{KL}}$, we define*

$$\sup_{x \sim d_0, a \in \mathcal{A}} \frac{\pi(a|x)}{\pi_0(a|x)} := C_{\rho_{\mathrm{KL}}}.$$

**Remark 2.9** (Relation between Local and Global Coverage Conditions). *The local-coverage condition (Definition 2.8) is more precise because compared to the global conditions targeting all possible policies, it only constrains the coverage to a KL-ball. In Song et al. (2024), because of the specific form of the oracle (Definition 2.5), the considered policy class is $\Pi = \{\pi(\cdot|\cdot) \propto \pi_0(\cdot|\cdot)\exp(\eta R(\theta, \cdot, \cdot)) : R(\theta, \cdot, \cdot) \in \mathcal{R}\}$. Thus, they only need to assume that the condition hold for $\rho = 2\eta B$, indicating that $C_{\rho_{\mathrm{KL}}} \leq C_{\mathrm{GL}}$. On the other hand, the data coverage condition (Definition 2.6) is measured on the level of reward functions instead of policies. In this sense, the data coverage condition and local-coverage condition do not encompass each other.*

## 3 KL-REGULARIZED CONTEXTUAL BANDITS

### 3.1 LOWER BOUND

In this section, we provide a lower bound for the KL-regularized contextual bandit problem.

**Theorem 3.1.** *For any $\epsilon \in (0, 1), \eta > 0$, and any algorithm $A$, there exists a KL-regularized contextual bandit problem with $O(1)$ coverage coefficient and reward function class $\mathcal{R}$ such that $A$ requires at least $\Omega\big(\min(\frac{\eta \log N_{\mathcal{R}}(\epsilon)}{\epsilon}, \frac{\log N_{\mathcal{R}}(\epsilon)}{\epsilon^2})\big)$ rounds to achieve a suboptimality gap of $\epsilon$.*

**Remark 3.2.** *The lower bound in Theorem 3.1 indicates that the sample complexity of the KL-regularized contextual bandit problem is $\Omega(\eta \log N_{\mathcal{R}}(\epsilon)/\epsilon)$ when $\epsilon$ is sufficiently small. In our proof, the KL-regularization term shifts the local landscape of the objective function, which prevents us to directly apply the standard bandit analysis, and thus requires a novel analysis to derive the new lower bound. This $\Omega(\eta \log N_{\mathcal{R}}(\epsilon)/\epsilon)$ lower bound suggests that the KL-regularized contextual bandit problem enjoys a lower sample complexity compared to the standard contextual bandit.*

## 3.2 The Proposed Algorithm

---

**Algorithm 1** Two-Stage mixed-policy sampling

---

1: **Input:** $\eta$, $\epsilon$, $\pi_0$, $\Theta$.
    ▷ Use policy $\pi_0$ to achieve sufficient data coverage
2: **for** $i = 1, \ldots, m$ **do**
3:    Sample context $x_i^0 \sim d_0$ and action $a_i^0 \sim \pi_0(\cdot|x_i^0)$.
4:    Observe reward $r_i^0 = R(\theta_*, x_i^0, a_i^0) + \epsilon_i^0$, where $\epsilon_i^0$ is the random noise.
5: **end for**
6: Compute the least square estimate of the reward function based on $D_0 = \{(x_i^0, a_i^0, r_i^0)\}_{i=1}^m$:

$$\widehat{\theta}_0 \leftarrow \underset{\theta \in \Theta}{\arg\min} \sum_{i=1}^m (R(\theta, x_i^0, a_i^0) - r_i^0)^2.$$

7: Apply the planning oracle to compute $\pi_{\widehat{\theta}_0}^\eta(\cdot|\cdot) \propto \pi_0(\cdot|\cdot) \exp\big(\eta R(\widehat{\theta}_0, \cdot, \cdot)\big)$.
    ▷ Use policy $\pi_{\widehat{\theta}_0}^\eta$ to sample new responses
8: **for** $i = 1, \ldots, n$ **do**
9:    Sample context $x_i \sim d_0$ and action $a_i \sim \pi_{\widehat{\theta}_0}^\eta(\cdot|x_i)$.
10:    Observe reward $r_i = R(\theta_*, x_i, a_i) + \epsilon_i$, where $\epsilon_i$ is the random noise.
11: **end for**
12: Compute the least square estimate of the reward function using $\{(x_i, a_i, r_i)\}_{i=1}^n$ together with $D_0$:

$$\widehat{\theta} \leftarrow \underset{\theta \in \Theta}{\arg\min} \sum_{i=1}^m (R(\theta, x_i^0, a_i^0) - r_i^0)^2 + \sum_{i=1}^n (R(\theta, x_i, a_i) - r_i)^2.$$

13: **Output** $\pi_{\widehat{\theta}}^\eta(\cdot|\cdot) \propto \pi_0(\cdot|\cdot) \exp\big(\eta R(\widehat{\theta}, \cdot, \cdot)\big)$.

---

We present the algorithmic framework in Algorithm 1 for the KL-regularized contextual bandit problem, which serves as a theoretical model for online RLHF with absolute-rating feedback. The algorithm consists of two states:

- In the first stage, we sample $m$ contexts (prompts) and actions (answers) from the foundation model $\pi_0$ and observe the corresponding rewards (absolute ratings). These ratings can be regarded as noisy observations of the underlying reward function $R(\theta_*, x, a)$. In line 6, we compute an estimate of the reward function $\widehat{\theta}_0$ using least squares regression based on the collected data. In line 7, we apply the planning oracle to obtain the policy $\pi_{\widehat{\theta}_0}^\eta$ which maximizes the following KL-regularized estimated objective in Definition 2.5 with reward function $R(\theta, \cdot, \cdot) = R(\widehat{\theta}_0, \cdot, \cdot)$.

- In the second stage, we utilize the trained policy $\pi_{\widehat{\theta}_0}^\eta$ to sample $n$ contexts (prompts) and actions (responses). With the intermediate policy $\pi_{\widehat{\theta}_0}^\eta$, we can collect new data $\{(x_i, a_i, r_i)\}_{i=1}^n$ which is more aligned with the data distribution induced by the optimal policy $\pi_*$. In line 12, the algorithm combines data from both stages $\{(x_i, a_i, r_i)\}_{i=1}^n$ and $\{(x_i^0, a_i^0, r_i^0)\}_{i=1}^m$ to compute a refined least squares estimate $\widehat{\theta}$ of the reward function, minimizing the sum of squared errors across both datasets. By aggregating the two datasets together, there is an overlap between the data to compute $\widehat{\theta}$ and $\widehat{\theta}_0$, so that the output policy $\pi_{\widehat{\theta}}^\eta$ is well covered by the intermediate policy $\pi_{\widehat{\theta}_0}^\eta$.

## 3.3 Theoretical Guarantees

**Loose Bound of Previous Analysis.** The previous analysis is loose since they basically follow the techniques of bandits and neglect the significance of KL-regularization. For simplicity, We use short-hand notation $R(\theta, x, \pi) = \mathbb{E}_{a \sim \pi(\cdot|x)} R(\theta, x, a)$ and denote $\mathrm{KL}(\pi(\cdot|x)\|\pi'(\cdot|x))$ by $\mathrm{KL}(\pi\|\pi')$ when there is no confusion. We make the estimation on a dataset $\{(x_i, a_i, r_i) : x_i \sim d_0, s_i \sim \pi_0\}_{i=1}^n$: $\pi_{\widehat{\theta}}^\eta = \arg\max_{\pi \in \Pi} \mathbb{E}_{x \sim d_0}[R(\widehat{\theta}, x, \pi) - \eta^{-1}\mathrm{KL}(\pi\|\pi_0)]$, and has a small in-sample-error: $\mathbb{E}_{x \sim d_0}\mathbb{E}_{a \sim \pi_o(\cdot|x)}\big[(R(\widehat{\theta}, x, a) - R(\theta^*, x, a))^2\big] = O(1/n)$. The sub-optimality is decomposed as:

$$Q(\pi^*) - Q(\pi_{\widehat{\theta}}^\eta) = \mathbb{E}_{x \sim d_0}\big[R(\theta^*, x, \pi^*) - R(\widehat{\theta}, x, \pi^*)\big] + \mathbb{E}_{x \sim d_0}\big[R(\widehat{\theta}, x, \pi_{\widehat{\theta}}^\eta) - R(\theta^*, x, \pi_{\widehat{\theta}}^\eta)\big]$$

$$+ \mathbb{E}_{x \sim d_0} \left[ R(\widehat{\theta}, x, \pi^*) - \eta^{-1} \mathrm{KL}(\pi^* \| \pi_0) \right] - \mathbb{E}_{x \sim d_0} \left[ R(\widehat{\theta}, x, \pi_{\widehat{\theta}}^{\eta}) - \eta^{-1} \mathrm{KL}(\pi_{\widehat{\theta}}^{\eta} \| \pi_0)) \right]$$

$$\leq \mathbb{E}_{x \sim d_0} \left[ R(\theta^*, x, \pi^*) - R(\widehat{\theta}, x, \pi^*) + R(\widehat{\theta}, x, \pi_{\widehat{\theta}}^{\eta}) - R(\theta^*, x, \pi_{\widehat{\theta}}^{\eta}) \right],$$

where the inequality holds since $\pi_{\widehat{\theta}}^{\eta}$ is the maximum.

Then, the suboptimality can be further bounded by using the coverage condition (Definition 2.7) and concentration inequalities:

$$Q(\pi^*) - Q(\pi_{\widehat{\theta}}^{\eta}) \leq 2 C_{\mathrm{GL}} \mathbb{E}_{x \sim d_0} \mathbb{E}_{a \sim \pi_0(\cdot|x)} \left[ |R(\theta^*, x, a) - R(\widehat{\theta}, x, a)| \right]$$

$$\leq 2 C_{\mathrm{GL}} \sqrt{\mathbb{E}_{a \sim \pi_0(\cdot|x)} \left[ (R(\theta^*, x, a) - R(\widehat{\theta}, x, a))^2 \right]} = O(C_{\mathrm{GL}} / \sqrt{n}).$$

Hence, they need $\Theta(C_{\mathrm{GL}}^2 / \epsilon^2)$ sample complexity to ensure $O(\epsilon)$ sub-optimility.

**Power of KL-regularization** The crucial point of the sharper result is utilizing the strong convexity of the objective $Q$ because of the KL-regularization. Specifically, we take the first-order Taylor expansion of sub-optimality with respect to $\{ \Delta(x, a) = R(\widehat{\theta}, x, a) - R(\theta^*, x, a) : a \in \mathcal{A} \}$

$$Q(\pi^*) - Q(\pi_{\widehat{\theta}}^{\eta}) = \eta \mathbb{E}_{x \sim d_0} \left[ \sum_{a \in \mathcal{A}} \pi_f^{\eta}(a|x) \Delta^2(x, a) - \sum_{a_1, a_2 \in \mathcal{A}} \pi_f^{\eta}(a_1|x) \pi_f^{\eta}(a_2|x) \Delta(x, a_1) \Delta(x, a_2) \right]$$

$$\leq \eta \mathbb{E}_{x \sim d_0} \left[ \sum_{a \in \mathcal{A}} \pi_f^{\eta}(a|x) \Delta^2(x, a) \right],$$

where $f(\cdot, \cdot) = \gamma R(\widehat{\theta}, \cdot, \cdot) + (1 - \gamma) R(\theta_*, \cdot, \cdot)$ ($\gamma \in (0, 1)$) the inequality uses the fact that second term on the right-hand side of the equality is $(\sum_{a \in \mathcal{A}} \pi_f^{\eta}(a|x) \Delta(x, a))^2 \geq 0$.

Now, under Algorithm 1, the coverage condition (Definition 2.6) and with concentration inequalities, if the datasize $m = \Theta(\eta^2 D^2)$, we can prove that for $\| R(\widehat{\theta}, \cdot, \cdot) - R(\theta^*, \cdot, \cdot) \|_\infty \leq \eta^{-1}$ and $\| R(\widehat{\theta}_0, \cdot, \cdot) - R(\theta^*, \cdot, \cdot) \|_\infty \leq \eta^{-1}$, which implies the whole-policy coverage condition: $\| \pi_f^{\eta}(\cdot|\cdot) / \pi_{\widehat{\theta}_0}^{\eta}(\cdot|\cdot) \|_\infty \leq e^4$. Therefore, by setting $n = \Theta(\eta/\epsilon)$, we obtain that $\pi_{\widehat{\theta}}^{\eta}$ is $O(\epsilon)$ optimal. The conclusion is presented in the following theorem.

**Theorem 3.3.** *Suppose that Assumption 2.6 holds. For any $\delta \in (0, 1/5)$, $\epsilon > 0$ and constant $c_{m,n} > 0$, if we set $m = \Theta(\eta^2 D^2 \cdot B^2 \log(2 N_{\mathcal{R}}(\epsilon_c)/\delta))$ and $n = \Theta(\eta/\epsilon \cdot B^2 \log(N_{\mathcal{R}}(\epsilon_c)/\delta))$ and $\epsilon_c = \min\{ \frac{\epsilon}{2(1 + c_{m,n}^{-1})B}, \frac{1}{8(1 + c_{m,n})B\eta^2 D^2} \}$, then with probability at least $1 - 5\delta$ the output policy of Algorithm 1 $\pi_{\widehat{\theta}}^{\eta}$ is $\mathcal{O}(\epsilon)$ optimal.*

**Remark 3.4.** *Theorem 3.3 shows that the sample complexity of Algorithm 1 is $\mathcal{O}(\eta/\epsilon \log N_{\mathcal{R}}(\epsilon/\delta))$ when the reward scale is a constant and $\epsilon$ is sufficiently small. The result indicates that the proposed two-stage mixed sampling strategy can achieve a suboptimality gap of $\epsilon$ with only an additive dependence on the coverage coefficient $D^2$.*

### 3.4 DISCUSSION: RESULT FOR LOCAL-COVERAGE

In this subsection, we consider a more general assumption as described in Definition 2.8.

**Corollary 3.5.** *Let $C_{\rho_{\mathrm{KL}}}$ be in Definition 2.8 where $\rho_{\mathrm{KL}} = 2\eta B$. For any $\delta \in (0, 1/6)$ and $\epsilon > 0$, if we set $n = c_{m,n} m = \Theta(C_{\rho_{\mathrm{KL}}} \eta/\epsilon \cdot B \log(N_{\mathcal{R}}(\epsilon_c)/\delta))$ (where constant $c_{m,n} > 0$, $\epsilon_c = \epsilon/(2(1 + c_{m,n}^{-1})B))$ then with probability at least $1 - 6\delta$ the output policy of Algorithm 2 $\pi_{\widehat{\theta}}^{\eta}$ is $O(\epsilon)$ optimal.*

In comparison with the sample complexity $\Theta(\eta^2 D^2 + \eta/\epsilon)$ under global-coverage in Theorem 3.3, the order $\Theta(C_{\rho_{\mathrm{KL}}} \eta/\epsilon)$ depends on a weaker coverage $C_{\rho_{\mathrm{KL}}}$, but only has a multiplicative dependence on the coverage coefficient instead of additive dependence. whether the additive dependence can be achieved under the local-coverage condition is left as future work. Moreover, we compare this result with Theorem 4.2 in Song et al. (2024) and suppose that the in-sample-error $\epsilon_{\mathrm{reward}}$ of Song et al. (2024) is $O(1/n)$, their sample complexity is $\Theta(C_{\rho_{\mathrm{KL}}}^2 / \epsilon^2)$, which is looser than ours $\Theta(C_{\rho_{\mathrm{KL}}} \eta/\epsilon)$ when $\eta = o(C_{\rho_{\mathrm{KL}}} / \epsilon)$.

## 4 REINFORCEMENT LEARNING FROM PREFERENCE FEEDBACK

In this section, we consider the problem of aligning the language model with preference feedback. As discussed in Section 2.2, at each round, we can sample a pair of actions (responses) $a_1, a_2$ and call

a preference oracle to get the preference label $y \in \{0, 1\}$, where $y = 1$ means that the user prefers $a_1$ over $a_2$ (Definition 2.2). To learn the reward function, we introduce the following assumption for step 1 to ensure the existence of an MLE estimation oracle that can globally maximize the likelihood of the BT model over all possible reward functions.

**Definition 4.1** (MLE estimation oracle). *Given a set of context-action pairs $\{(x_i, a_i^1, a_i^2, y_i)\}_{i=1}^n$ generated from the BT model, can output the parameter $\widehat{\theta}$ such that*

$$\widehat{\theta} = \arg\max_{\theta \in \Theta} \sum_{i=1}^n \underbrace{y_i \cdot \log \sigma(R(\theta, x_i, a_i^1) - R(\theta, x_i, a_i^2)) + (1 - y_i) \cdot \log \sigma(R(\theta, x_i, a_i^2) - R(\theta, x_i, a_i^1))}_{\mathcal{L}(\theta | x_i, a_i^1, a_i^2, y_i)}.$$

Following the previous analysis for RLHF (Xiong et al., 2024a), we assume the existence of a policy improvement oracle (Definition 2.5, corresponding to step 2) that can compute the optimal policy $\pi_{\widehat{\theta}}^\eta$ based on the reward function $\widehat{\theta}$.

**Remark 4.2.** *We learn the reward function since we can always control the reward (like clipping and normalization) to ensure that the reward function is always bounded by $B$. The bounded assumption does not apply for direct preference learning like DPO (Rafailov et al., 2024) since there is no intrinsic policy function class encompassing the soundness (Song et al., 2024), thus increasing the cases of overfitting.*

### 4.1 THEORETICAL GUARANTEES

**Lower Bound** We provide a lower bound for the RLHF problem with preference feedback. The lower bound is derived by constructing a hard instance where the reward function is difficult to estimate from the preference feedback.

**Theorem 4.3.** *For any $\epsilon \in (0, 1), \eta > 0$, and any algorithm A, there exists a KL-regularized preference learning problem as defined in Section 2.2 with $O(1)$ coverage coefficient and reward function class $\mathcal{R}$ such that A requires at least $\Omega\big(\min(\frac{\eta \log N_\mathcal{R}(\epsilon)}{\epsilon}, \frac{\log N_\mathcal{R}(\epsilon)}{\epsilon^2})\big)$ samples to achieve a suboptimality gap of $\epsilon$.*

We defer Algorithm 2, a 2-stage mixed-policy sampling algorithm for RLHF with preference feedback, to Appendix A for conciseness because of its similarity to Algorithm 1.

**Upper bound for global coverage** We provide the theoretical guarantees for Algorithm 2 in the following theorem.

**Theorem 4.4.** *Suppose that Assumption 2.6 holds. For any $\delta \in (0, 1/6)$ and $\epsilon > 0$, if we set $m = \Theta(\eta^2 D^2 \cdot e^B \log(N_\mathcal{R}(\epsilon_c)/\delta))$ and $n = \Theta(\eta/\epsilon \cdot e^B \log(N_\mathcal{R}(\epsilon_c)/\delta))$ (where $\epsilon_c = \min\{\frac{\epsilon}{2(1+c_{m,n}^{-1})e^B}, \frac{1}{8(1+c_{m,n})e^B \eta^2 D^2}\}$) then with probability at least $1 - 6\delta$ the output policy of Algorithm 2 $\pi_{\widehat{\theta}}^\eta$ is $O(\epsilon)$ optimal.*

**Remark 4.5** (Comparison with Hybrid Framework). *We compare our two-stage mixed sampling method with hybrid frameworks. From the algorithmic perspective, a hybrid algorithm first learns from an offline dataset and then requires sufficient online iterations to ensure the performance (Xiong et al., 2024a). For example, for a finite action space with A actions, the number of online iterations should be $\Theta(A)$. In contrast, our method only requires two iterations of sampling from mixed policy and interacting with the environment. Moreover, the results of hybrid literature depend on both the coverage coefficient and the structure complexity of the function class (like the dimension for a linear function class or eluder dimension (Russo & Van Roy, 2013)). Our result only needs the coverage condition of the reference policy. More importantly, we obtain a sharper bound on the sample complexity and derive the additive dependence on the coverage coefficient.*

**Remark 4.6.** *Although the coefficient $e^B$ appearing in sample size $m, n$ can be exponentially large, this term is caused by the non-linearity of the link function for the preference model, and is common in RLHF literature (Zhu et al., 2023; Xiong et al., 2024a; Ye et al., 2024b; Song et al., 2024).*

Theorem 4.4 shows that the sample complexity of Algorithm 2 is $\mathcal{O}(\eta/\epsilon \log N_\mathcal{R}(\epsilon/\delta))$ when the reward scale is a constant and $\epsilon$ is sufficiently small. The result indicates that the proposed two-stage mixed sampling strategy can achieve a suboptimality gap of $\epsilon$ with only an additive dependence on the coverage coefficient $D^2$.

Besides, the algorithm only requires sampling from the reference policy $\pi_0$ and the intermediate policy $\pi_{\widehat{\theta}_0}^\eta$, which is more aligned with the practical scenarios where the preference feedback is

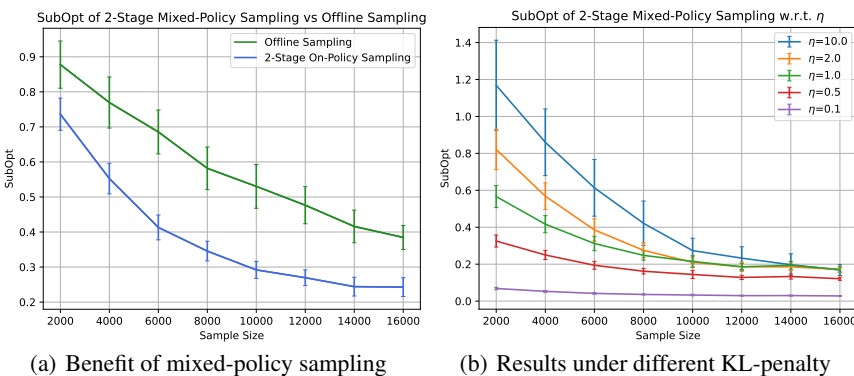

(a) Benefit of mixed-policy sampling      (b) Results under different KL-penalty

Figure 1: Suboptimality gap for reinforcement learning from preference feedback.

collected from the human users and it is expensive to collect the data while the language model is being updated. Our result implies that we may achieve a near-optimal sample complexity by simply leveraging an intermediate policy to collect more data, and the training process of the reward model and the policy (language model) can be highly decoupled.

**Upper bound for local coverage**   We also show the result under the local-coverage assumption (Definition 2.8) as follows.

**Corollary 4.7.** *Let $C_{\rho_{\mathrm{KL}}}$ be in Definition 2.8 where $\rho = 2\eta B$. For any $\delta \in (0, 1/6)$ and $\epsilon > 0$, if we set $n = c_{m,n}m = \Theta(C_{\rho_{\mathrm{KL}}}\eta/\epsilon \cdot e^B \log(N_{\mathcal{R}}(\epsilon_c)/\delta))$ (where constant $c_{m,n} > 0$, $\epsilon_c = \frac{\epsilon}{2(1+c_{m,n}^{-1})e^B}$) then with probability at least $1 - 6\delta$ the output policy of Algorithm 2 $\pi_{\hat{\theta}}^\eta$ is $O(\epsilon)$ optimal.*

## 5   Experimental Results

In this section, we conduct experiments with synthetic data to investigate the benefit of mixed-policy sampling and the effect of KL-regularization coefficient on the sample complexity of the problem. We plot the experimental results for RL from preference feedback in Figure 1 and defer the results for KL-regularized contextual bandits in Appendix B. All of the trials are repeated for 10 times and plotted with the standard variation.

We consider the case where context distribution $d_0$ is a projected gaussian distribution over the unit sphere and $\mathcal{A}$ is a discrete set with $|\mathcal{A}| = 5$. We construct the reward functions as $R(\phi, x, a) = \langle x, \phi(a) \rangle$, parameterized by a mapping $\phi$ from $\mathcal{A}$ to $\mathbb{R}^{10}$. To generate $\phi_*$, we sample $\phi_*(a)$ independently for each $a \in \mathcal{A}$ according to another projected gaussian distribution over the sphere with radius equal to 5. In Figure 1(a), we compare the suboptimality gaps of mixed-policy sampling with $m = n$ to those of offline sampling using $\pi_0$ under the same sample sizes. The result indicates that the usage of mixed-policy sampling reduces the suboptimality gap by a large margin. In Figure 1(b), it is shown that the sample complexity is remarkably affected by the KL-regularization term, corroborating our sharp analysis for regularized RLHF.

## 6   Conclusion

We have presented a comprehensive theoretical analysis of the role of reverse-KL regularization in decision-making models including contextual bandits and reinforcement learning from preference feedback, highlighting its significance in terms of sample complexity. Our results provide new insights into the power of regularization extending beyond its traditional role of mitigating errors from the current critic (or reward) model.

Additionally, we examined the role of data coverage in both contextual bandits and RLHF. Our analysis shows that with sufficient coverage from the reference policy, a mixed sampling strategy can achieve a sample complexity that exhibits only an additive dependence on the coverage coefficient without the need for explicit exploration or additional structural assumptions.

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

## A  ALGORITHM FOR PREFERENCE FEEDBACK

---

**Algorithm 2** 2-Stage mixed-policy sampling for preference feedback

---

1: **Input:** $\eta, \epsilon, \pi_0, \Theta$.
   ▷ Use policy $\pi_0$ to achieve sufficient data coverage
2: **for** $i = 1, \ldots, m$ **do**
3:     Sample context $\widetilde{x}_i \sim d_0$ and 2 actions $\widetilde{a}_i^1, \widetilde{a}_i^2 \sim \pi_0(\cdot|\widetilde{x}_i)$.
4:     Observe preference label $\widetilde{y}_i \in \{0, 1\}$ from the preference oracle defined in Definition 2.2.
5: **end for**
6: Compute the MLE estimator of the reward function based on $\{(\widetilde{x}_i, \widetilde{a}_i^1, \widetilde{a}_i^2, \widetilde{y}_i)\}_{i=1}^m$:

$$\widehat{\theta}_0 \leftarrow \arg\max_\theta \sum_{i=1}^m \widetilde{y}_i \cdot \log \sigma(R(\theta, \widetilde{x}_i, \widetilde{a}_i^1) - R(\theta, \widetilde{x}_i, \widetilde{a}_i^2)) + (1 - \widetilde{y}_i) \cdot \log \sigma(R(\theta, \widetilde{x}_i, \widetilde{a}_i^2) - R(\theta, \widetilde{x}_i, \widetilde{a}_i^1)).$$

7: Apply the planning oracle to compute $\pi_{\widehat{\theta}_0}^\eta(\cdot|\cdot) \propto \pi_0(\cdot|\cdot) \exp(\eta R(\widehat{\theta}_0, \cdot, \cdot))$.
   ▷ Use policy $\pi_{\widehat{\theta}_0}^\eta$ to sample new responses
8: **for** $i = 1, \ldots, n$ **do**
9:     Sample context $x_i \sim d_0$ and 2 actions $a_i^1, a_i^2 \sim \pi_{\widehat{\theta}_0}^\eta(\cdot|x_i)$.
10:     Observe preference label $y_i \in \{0, 1\}$ from the preference oracle defined in Definition 2.2.
11: **end for**
12: Compute the MLE estimator of the reward function using $\{(x_i, a_i^1, a_i^2, y_i)\}_{i=1}^n$ together with $\{(\widetilde{x}_i, \widetilde{a}_i^1, \widetilde{a}_i^2, \widetilde{y}_i)\}_{i=1}^m$:

$$\widehat{\theta} \leftarrow \arg\max_\theta \sum_{i=1}^m \widetilde{y}_i \cdot \log \sigma(R(\theta, \widetilde{x}_i, \widetilde{a}_i^1) - R(\theta, \widetilde{x}_i, \widetilde{a}_i^2)) + (1 - \widetilde{y}_i) \cdot \log \sigma(R(\theta, \widetilde{x}_i, \widetilde{a}_i^2) - R(\theta, \widetilde{x}_i, \widetilde{a}_i^1))$$

$$+ \sum_{i=1}^n y_i \cdot \log \sigma(R(\theta, x_i, a_i^1) - R(\theta, x_i, a_i^2)) + (1 - y_i) \cdot \log \sigma(R(\theta, x_i, a_i^2) - R(\theta, x_i, a_i^1))$$

13: **Output** $\pi_{\widehat{\theta}}^\eta(\cdot|\cdot) \propto \pi_0(\cdot|\cdot) \exp(\eta R(\widehat{\theta}, \cdot, \cdot))$.

---

In the first stage, we sample $m$ context-action pairs $\{(\widetilde{x}_i, \widetilde{a}_i^1, \widetilde{a}_i^2, \widetilde{y}_i)\}_{i=1}^m$ from the BT model and call the preference oracle to get the preference labels. We then compute the MLE estimator of the reward function $\widehat{\theta}_0$ based on the preference feedback in line 6. Afterwards, we apply the planning oracle to compute the optimal policy $\pi_{\widehat{\theta}_0}^\eta$ based on the reward function $\widehat{\theta}_0$ in line 7. Line 6 and line 7 correspond to the practical implementation of RLHF(Ouyang et al., 2022; Bai et al., 2022; Touvron et al., 2023) given a dataset of preference feedback.

In the second stage, we sample $n$ context-action pairs $\{(x_i, a_i^1, a_i^2, y_i)\}_{i=1}^n$ using the intermediate policy $\pi_{\widehat{\theta}_0}^\eta$ and call the preference oracle to get the preference labels. We then compute the MLE estimator of the reward function $\widehat{\theta}$ based on the preference feedback from both stages. Finally, we apply the planning oracle to compute the optimal policy $\pi_{\widehat{\theta}}^\eta$ based on the reward function $\widehat{\theta}$.

## B  ADDITIONAL EXPERIMENTAL RESULTS

We show our experimental results for regularized contextual bandits in the following figure 2, which also corroborate our theory.

## C  PROOFS FROM SECTION 3

### C.1  PROOF OF THEOREM 3.1

*Proof of Theorem 3.1.* Consider a simple case when $|\mathcal{X}| = M$ and $|\mathcal{A}| = 2$. We suppose that the context $x$ is drawn uniformly from $\mathcal{X}$ at the beginning of each round. Let $\Theta$ be the set consisting of

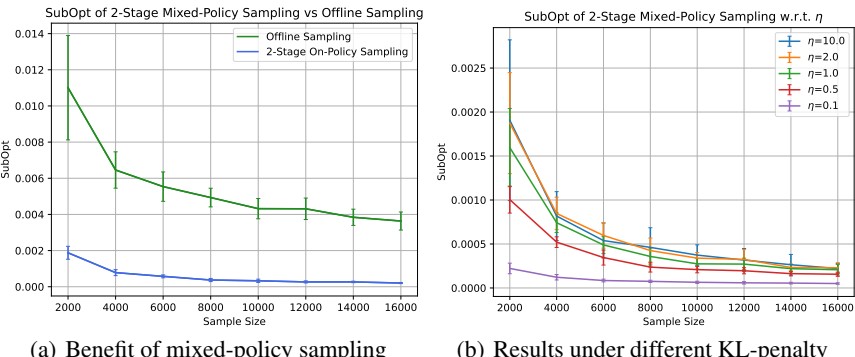

Figure 2: Suboptimality gap for KL-regularized contextual bandits.

mappings from $\mathcal{X}$ to $\mathcal{A} = \{0, 1\}$. For each $\theta \in \Theta$, we have $R(\theta, x, a) = \begin{cases} 1/2 + c & \text{if } a = \theta(x), \\ 1/2 & \text{if } a \neq \theta(x), \end{cases}$

where $c > 0$ is a constant, and $\theta(x)$ is the optimal action under context $x$ when the model is $\theta$.

For any $(\theta, x, a) \in \Theta \times \mathcal{X} \times \mathcal{A}$, we assume the reward feedback $r \sim Bernoulli(R(\theta, x, a))$ when the model is $\theta$ and $a$ is chosen under context $x$.

We pick a pair of model $\theta_1, \theta_2$ in $\Theta$, such that $\theta_1(x) = \begin{cases} \theta_2(x) & \text{if } x \neq x_0, \\ 1 - \theta_2(x) & \text{if } x = x_0. \end{cases}$

We denote by $\mathbb{P}_\theta, \mathbb{E}_\theta$ the probability measure and expectation under the model $\theta$.

Applying Pinsker's inequality (Lemma E.3), we have for all event $A$ measurable with respect to the filtration generated by the observations,

$$|\mathbb{P}_{\theta_1}(A) - \mathbb{P}_{\theta_2}(A)| \leq \sqrt{\frac{1}{2} \log(1 - 4c^2) \mathbb{E}_{\theta_1}[N(x_0)]} \leq \sqrt{2c^2 \mathbb{E}_{\theta_1}[N(x_0)]} = \sqrt{2c^2 T/M},$$

where the first inequality follows from the chain rule of KL divergence, and the fact that $\mathbb{E}_{\theta_1}[N(x_0)] = T/M$.

Set $A$ to be the event that $\pi_{out}(\theta_1(x_0)|x_0) > 1/2$. Then we have

$$\mathbb{P}_{\theta_1}(\pi_{out}(\theta_1(x_0)|x_0) \leq 1/2) + \mathbb{P}_{\theta_2}(\pi_{out}(\theta_2(x_0)|x_0) \leq 1/2) \geq 1 - |\mathbb{P}_{\theta_1}(A) - \mathbb{P}_{\theta_2}(A)| \geq 1 - \sqrt{2c^2 T/M}.$$

If the model $\theta$ is uniformly drawn from $\Theta$, then we have

$$\mathbb{E}_{\theta \sim Unif(\Theta)} \mathbb{P}_\theta(\pi_{out}(\theta(x_0)) \leq 1/2) \geq \frac{1}{2} - \sqrt{c^2 T/2M}$$

for an arbitrary $x_0$.

Then we consider the following suboptimality gap:

$$\mathbb{E}_{\pi_{\theta_*}^\eta} \left[ R(\theta_*, x, a) - \frac{1}{\eta} \ln \frac{\pi_{\theta_*}^\eta(a|x)}{\pi_0(a|x)} \right] - \mathbb{E}_{\pi_{out}} \left[ R(\theta_*, x, a) - \frac{1}{\eta} \ln \frac{\pi_{out}(a|x)}{\pi_0(a|x)} \right]$$

$$= \frac{1}{\eta} \mathbb{E}_{\pi_{\theta_*}^\eta} \left[ \ln \frac{\pi_0(a|x) \cdot \exp(\eta R(\theta_*, x, a))}{\pi_{\theta_*}^\eta(a|x)} \right] - \frac{1}{\eta} \mathbb{E}_{\pi_{out}} \left[ \ln \frac{\pi_0(a|x) \cdot \exp(\eta R(\theta_*, x, a))}{\pi_{out}(a|x)} \right]$$

$$= \frac{1}{\eta} \mathbb{E}_{\pi_{out}} \left[ \ln \frac{\pi_{out}(a|x)}{\pi^*(a|x)} \right],$$

where the last equality follows from the fact that $\pi_{\theta_*}^\eta \propto \pi_0(a|x) \cdot \exp(\eta R(\theta_*, x, a))$.

To bound the suboptimality gap, we further have

$$\mathbb{E}_{\theta \sim Unif(\Theta)} \mathbb{E}_{\pi_{out}} \left[ \ln \frac{\pi_{out}(a|x)}{\pi^*(a|x)} \right]$$

$$= \mathbb{E}_{\theta \sim Unif(\Theta)} \frac{1}{M} \sum_{x \in \mathcal{X}} \mathbb{E}_{a \sim \pi_{out}(\cdot|x)} \left[ \ln \frac{\pi_{out}(a|x)}{\pi^*(a|x)} \right]$$

$$\geq \mathbb{E}_{\theta \sim Unif(\Theta)} \frac{1}{M} \sum_{x \in \mathcal{X}} \mathbb{P}_\theta \left( \pi_{out}(\theta(x)) \leq 1/2 \right) \cdot \left[ \frac{1}{2} \ln \frac{1 + \exp(-\eta c)}{2} + \frac{1}{2} \ln \frac{1 + \exp(\eta c)}{2} \right]$$

$$\geq \left( \frac{1}{2} - \sqrt{c^2 T/2M} \right) \left[ \frac{1}{2} \ln \frac{1 + \exp(-\eta c)}{2} + \frac{1}{2} \ln \frac{1 + \exp(\eta c)}{2} \right] \tag{C.1}$$

Note that

$$\frac{\mathrm{d}}{\mathrm{d}u} \left[ \frac{1}{2} \ln \frac{1 + e^{-u}}{2} + \frac{1}{2} \ln \frac{1 + e^u}{2} \right] \bigg|_{u=0} = \frac{1}{2} \left[ \frac{1}{1 + \exp(-u)} - \frac{1}{1 + \exp(u)} \right] \bigg|_{u=0} = 0,$$

$$\frac{\mathrm{d}^2}{\mathrm{d}u^2} \left[ \frac{1}{2} \ln \frac{1 + e^{-u}}{2} + \frac{1}{2} \ln \frac{1 + e^u}{2} \right] = \frac{\exp(u)}{[1 + \exp(u)]^2}.$$

Thus, applying Taylor's expansion on the right-hand side of (C.1), we have

$$\mathbb{E}_{\theta \sim Unif(\Theta)} \mathbb{E}_{\pi_{out}} \left[ \ln \frac{\pi_{out}(a|x)}{\pi^*(a|x)} \right] \geq \frac{1}{2} \cdot \left( \frac{1}{2} - \sqrt{c^2 T/2M} \right) \eta^2 c^2 \cdot \frac{1}{3 + \exp(\eta c)}$$

When $\epsilon < 1/64\eta$, we can set $c = 8\sqrt{\epsilon/\eta}$. To achieve a suboptimality gap of $\epsilon$, we need to satisfy:

$$\frac{1}{2} \cdot \left( \frac{1}{2} - \sqrt{c^2 T/2M} \right) \eta^2 c^2 \cdot \frac{1}{3 + \exp(\eta c)} \leq \eta \epsilon,$$

indicating that $T \geq \frac{\eta M}{512\epsilon} = \Omega(\frac{\eta M}{\epsilon})$.

When $\epsilon \geq 1/64\eta$, or equivalently, $\eta \geq 1/64\epsilon$, we employ a different lower bound for (C.1) as follows:

$$\frac{1}{2} \ln \frac{1 + \exp(-\eta c)}{2} + \frac{1}{2} \ln \frac{1 + \exp(\eta c)}{2} = \frac{1}{2} \ln \frac{2 + \exp(\eta c) + \exp(-\eta c)}{4}$$

$$\geq \frac{1}{2} \cdot \frac{1}{2} \left( \ln \frac{\exp(\eta c) + \exp(-\eta c)}{2} \right)$$

$$\geq \frac{1}{4} (\eta c - \ln 2), \tag{C.2}$$

where the first inequality follows from Jensen's inequality.

Substituting (C.2) into (C.1), we have

$$\epsilon \geq \frac{1}{\eta} \mathbb{E}_{\theta \sim Unif(\Theta)} \mathbb{E}_{\pi_{out}} \left[ \ln \frac{\pi_{out}(a|x)}{\pi^*(a|x)} \right] \geq \frac{1}{4} \cdot \left( \frac{1}{2} - \sqrt{c^2 T/2M} \right) (\eta c - \ln 2) \cdot \frac{1}{\eta}.$$

Set $c = 64\epsilon$. Then we have $T = \Omega(M/\epsilon^2)$.

$\square$

## C.2 PROOF OF THEOREM 3.3

We start with the following lemma, which provides an on-policy generalization bound for the reward function. Due to the on-policy nature of the algorithm (i.e., the usage of intermediate $\pi_{\hat{\theta}_0}^\eta$), we can leverage the covering number of the reward function class $\mathcal{R}$ to derive the generalization error. Since we are using a fixed policy $\pi_{\hat{\theta}_0}^\eta$ to sample in the second stage, we can derive the generalization error of the reward function as follows:

**Lemma C.1** (Generalization error of reward function). *For an arbitrary policy $\pi$, a set of context-action pairs $\{(x_i, a_i)\}_{i=1}^n$ generated i.i.d. from $\pi$, and a distance threshold $0 < \epsilon_c \leq B$, we have with probability at least $1 - \delta$, for any pair of parameters $\theta_1$ and $\theta_2$,*

$$\mathbb{E}_\pi |R(\theta_1, x, a) - R(\theta_2, x, a)|^2$$

$$\leq \frac{2}{n} \sum_{i=1}^n |R(\theta_1, x_i, a_i) - R(\theta_2, x_i, a_i)|^2 + \frac{32B^2}{3n} \log(2N_\mathcal{R}(\epsilon_c)/\delta) + 10\epsilon_c B.$$

*Proof.* We first consider an $\epsilon_c$-net $\mathcal{R}^c$ of the reward function class $\mathcal{R}$ where $\mathcal{R}^c = \{R(\theta, \cdot, \cdot) | \theta \in \Theta^c\}$ with size $N_{\mathcal{R}}(\epsilon_c)$. For any $R(\theta, \cdot, \cdot) \in \mathcal{R}$, there exists $\theta^c$ such that $\|R(\theta, \cdot, \cdot) - R(\theta^c, \cdot, \cdot)\|_\infty \leq \epsilon_c$.

By Lemma E.1, for each pair of $\theta_1^c, \theta_2^c \in \Theta^c$ (corresponding to $\theta_1, \theta_2$), we have with probability at least $1 - \delta$,

$$\left| \frac{1}{n} \sum_{i=1}^n (R(\theta_1^c, x_i, a_i) - R(\theta_2^c, x_i, a_i))^2 - \mathbb{E}_\pi |R(\theta_1^c, x, a) - R(\theta_2^c, x, a)|^2 \right|$$

$$\leq \sqrt{\frac{2\mathrm{Var}_\pi |R(\theta_1^c, x, a) - R(\theta_2^c, x, a)|^2}{n} \log(2/\delta)} + \frac{2}{3n} B^2 \log(2/\delta)$$

$$\leq \sqrt{\frac{2B^2 \mathbb{E}_\pi |R(\theta_1^c, x, a) - R(\theta_2^c, x, a)|^2}{n} \log(2/\delta)} + \frac{2}{3n} B^2 \log(2/\delta)$$

where the second inequality follows from the fact that $R(\theta_1^c, x, a), R(\theta_2^c, x, a) \leq B$.

Using union bound over all $\theta_1^c, \theta_2^c \in \Theta^c$, we have with probability at least $1 - \delta$, for all $\theta_1^c, \theta_2^c \in \Theta^c$,

$$\mathbb{E}_\pi |R(\theta_1^c, x, a) - R(\theta_2^c, x, a)|^2 - \frac{1}{n} \sum_{i=1}^n (R(\theta_1^c, x_i, a_i) - R(\theta_2^c, x_i, a_i))^2$$

$$\leq \sqrt{\frac{4B^2 \mathbb{E}_\pi |R(\theta_1^c, x, a) - R(\theta_2^c, x, a)|^2}{n} \log(2N_{\mathcal{R}}(\epsilon_c)/\delta)} + \frac{4B^2}{3n} \log(2N_{\mathcal{R}}(\epsilon_c)/\delta),$$

from which we further obtain the following inequality by Lemma E.2,

$$\mathbb{E}_\pi |R(\theta_1^c, x, a) - R(\theta_2^c, x, a)|^2 \leq \frac{2}{n} \sum_{i=1}^n (R(\theta_1^c, x_i, a_i) - R(\theta_2^c, x_i, a_i))^2 + \frac{32B^2}{3n} \log(2N_{\mathcal{R}}(\epsilon_c)/\delta).$$

$$\tag{C.3}$$

Then we can complete the proof by the definition of $\epsilon$-net. $\qquad\square$

Next, we provide the following lemma, which gives an upper bound on the cumulative square error of the learned reward function.

**Lemma C.2** (Confidence bound for reward function). *For an arbitrary policy $\pi$, and a set of data $\{(x_i, a_i, r_i)\}_{i=1}^n$ generated i.i.d. from $\pi$, suppose that $\widehat{\theta}$ is the least squares estimator of $\theta_*$, i.e., $\widehat{\theta} = \arg\min_{\theta \in \Theta} \sum_{i=1}^n (R(\theta, x_i, a_i) - r_i)^2$. Then for any threshold $\epsilon_c > 0$, with probability at least $1 - \delta$, it holds that*

$$\sum_{i=1}^n (R(\widehat{\theta}, x_i, a_i) - R(\theta_*, x_i, a_i))^2 \leq 16B^2 \log(2N_{\mathcal{R}}(\epsilon_c)/\delta) + 4\epsilon_c n B.$$

*Proof.* We have the following inequality for $\sum_{i=1}^n (R(\widehat{\theta}, x_i, a_i) - R(\theta_*, x_i, a_i))^2$,

$$\sum_{i=1}^n (R(\widehat{\theta}, x_i, a_i) - R(\theta_*, x_i, a_i))^2$$

$$= \sum_{i=1}^n (R(\widehat{\theta}, x_i, a_i) - r_i)^2 - \sum_{i=1}^n (R(\theta_*, x_i, a_i) - r_i)^2$$

$$+ 2 \sum_{i=1}^n (R(\widehat{\theta}, x_i, a_i) - R(\theta_*, x_i, a_i))(r_i - R(\theta_*, x_i, a_i))$$

$$\leq 2 \sum_{i=1}^n (R(\widehat{\theta}, x_i, a_i) - R(\theta_*, x_i, a_i))(r_i - R(\theta_*, x_i, a_i)),$$

where the last inequality follows from the fact that $\sum_{i=1}^n (R(\widehat{\theta}, x_i, a_i) - r_i)^2 \leq \sum_{i=1}^n (R(\theta_*, x_i, a_i) - r_i)^2$.

We then consider an $\epsilon_c$-net $\mathcal{R}^c$ of the reward function class $\mathcal{R}$ where $\mathcal{R}^c = \{R(\theta, \cdot, \cdot) | \theta \in \Theta^c\}$ with size $N_{\mathcal{R}}(\epsilon_c)$. For any $R(\theta, \cdot, \cdot) \in \mathcal{R}$, there exists $\theta^c$ such that $\|R(\theta, x, a) - R(\theta^c, x, a)\|_\infty \le \epsilon_c$. From Azuma-Hoeffding inequality, with probability at least $1 - \delta$, it holds for all $\theta \in \Theta^c$ that

$$\sum_{i=1}^n (R(\theta, x_i, a_i) - R(\theta_*, x_i, a_i))(r_i - R(\theta_*, x_i, a_i))$$

$$\le \sqrt{2B^2 \sum_{i=1}^n (R(\theta, x_i, a_i) - R(\theta_*, x_i, a_i))^2 \log(2N_{\mathcal{R}}(\epsilon_c)/\delta)}.$$

Then we further have with probability at least $1 - \delta$, there exists $\|R(\theta^c, \cdot, \cdot) - R(\widehat{\theta}, \cdot, \cdot)\| \le \epsilon_c$ such that

$$\sum_{i=1}^n (R(\widehat{\theta}, x_i, a_i) - R(\theta_*, x_i, a_i))(r_i - R(\theta_*, x_i, a_i))$$

$$\le \sqrt{2B^2 \sum_{i=1}^n (R(\theta, x_i, a_i) - R(\theta_*, x_i, a_i))^2 \log(2N_{\mathcal{R}}(\epsilon_c)/\delta)} + 2\epsilon_c nB,$$

which implies that

$$\sum_{i=1}^n (R(\widehat{\theta}, x_i, a_i) - R(\theta_*, x_i, a_i))^2 \le 16B^2 \log(2N_{\mathcal{R}}(\epsilon_c)/\delta) + 4\epsilon_c nB \qquad (C.4)$$

from Lemma E.2. $\qquad \square$

With the above lemmas, we are now ready to prove the following lemma that bounds the estimation error of the reward function $R(\widehat{\theta}, \cdot, \cdot)$ under the sampled policy $\pi_{\widehat{\theta}_0}^\eta$.

**Lemma C.3.** *Let $\widehat{\theta}_0$ be the least squares estimator of the reward function based on the data $\{(x_i^0, a_i^0, r_i^0)\}_{i=1}^m$ generated from $\pi_0$ as defined in Algorithm 1. Then for any threshold $\epsilon_c > 0$, with probability at least $1 - 2\delta$, we have*

$$\mathbb{E}_{\pi_{\widehat{\theta}_0}^\eta} |R(\widehat{\theta}, x, a) - R(\theta_*, x, a)|^2 \le \frac{43B^2}{n} \log(2N_{\mathcal{R}}(\epsilon_c)/\delta) + 10\epsilon_c(1 + m/n)B.$$

*Proof.* By Lemma C.1, we have with probability at least $1 - \delta$, the following upper bound holds for $\mathbb{E}_{\pi_{\widehat{\theta}_0}^\eta} |R(\theta_1, x, a) - R(\theta_2, x, a)|^2$,

$$\mathbb{E}_{\pi_{\widehat{\theta}_0}^\eta} |R(\theta_1, x, a) - R(\theta_2, x, a)|^2$$

$$\le \frac{2}{n} \sum_{i=1}^n |R(\theta_1, x_i, a_i) - R(\theta_2, x_i, a_i)|^2 + \frac{32B^2}{3n} \log(2N_{\mathcal{R}}(\epsilon_c)/\delta) + 10\epsilon_c B. \qquad (C.5)$$

By Lemma C.2, with probability at least $1 - \delta$

$$\sum_{i=1}^n |R(\theta_*, x_i, a_i) - R(\widehat{\theta}, x_i, a_i)|^2 \le 16B^2 \log(2N_{\mathcal{R}}(\epsilon_c)/\delta) + 4\epsilon_c(n + m)B. \qquad (C.6)$$

Then we can complete the proof using a union bound and substituting (C.6) into (C.5). $\qquad \square$

**Lemma C.4.** *If $m \ge 128\eta^2 D^2 B^2 \cdot \log(2N_{\mathcal{R}}(\epsilon_c)/\delta))$, and there exists a positive constant $c_{m,n} > 0$ such that $n = c_{m,n}n$ in Algorithm 1 and Assumption 2.6 holds, then by taking $\epsilon_c \le \min\{B, (8(1 + c_{m,n})B\eta^2 D^2)^{-1}\}$, with probability at least $1 - 3\delta$, we have*

$$\eta|R(\widehat{\theta}_0, x, a) - R(\theta_*, x, a)| \le 1, \quad \eta|R(\widehat{\theta}, x, a) - R(\theta_*, x, a)| \le 1$$

*for any pair $(x, a) \in \mathcal{X} \times \mathcal{A}$ such that $\pi_0(a|x) > 0$.*

*Proof.* By Lemma C.1, with probability at least $1 - \delta$, for all $\theta_1, \theta_2 \in \Theta$, we have

$$\mathbb{E}_{\pi_0}|R(\theta_1, x, a) - R(\theta_2, x, a)|^2 \leq \frac{2}{m} \sum_{i=1}^{m} |R(\theta_1, x_i^0, a_i^0) - R(\theta_2, x_i^0, a_i^0)|^2 + \frac{32B^2}{3m} \log(2N_{\mathcal{R}}(\epsilon_c)/\delta).$$

By Lemma C.2, with probability at least $1 - \delta$, we have

$$\sum_{i=1}^{m} |R(\widehat{\theta}_0, x_i^0, a_i^0) - R(\theta_*, x_i^0, a_i^0)|^2 \leq 16B^2 \log(2N_{\mathcal{R}}(\epsilon_c)/\delta) + 4\epsilon_c m.$$

Also, with probability at least $1 - \delta$, we have

$$\sum_{i=1}^{m} |R(\theta_*, x_i^0, a_i^0) - R(\widehat{\theta}, x_i^0, a_i^0)|^2 \leq 16B^2 \log(2N_{\mathcal{R}}(\epsilon_c)/\delta) + 4\epsilon_c(m + n)B.$$

Similar to the proof of Lemma C.3, we have if $m \geq 128\eta^2 D^2 B^2 \cdot \log(2N_{\mathcal{R}}(\epsilon_c)/\delta)$, $n = c_{m,n}n$, then with probability at least $1 - 3\delta$,

$$\mathbb{E}_{\pi_0}|R(\theta_*, x, a) - R(\widehat{\theta}_0, x, a)|^2 \leq 1/\eta^2 D^2, \quad \mathbb{E}_{\pi_0}|R(\theta_*, x, a) - R(\widehat{\theta}, x, a)|^2 \leq 1/\eta^2 D^2.$$

which implies that $\eta|R(\widehat{\theta}_0, x, a) - R(\theta_*, x, a)| \leq 1$ and $\eta|R(\widehat{\theta}, x, a) - R(\theta_*, x, a)| \leq 1$ for all $(x, a) \in \mathcal{X} \times \mathcal{A}$ such that $\pi_0(a|x) > 0$. $\qquad\square$

*Proof of Theorem 3.3.* We have

$$\mathbb{E}_{\pi_{\theta_*}^\eta}\left[R(\theta_*, x, a) - \frac{1}{\eta}\ln\frac{\pi_{\theta_*}^\eta(a|x)}{\pi_0(a|x)}\right] - \mathbb{E}_{\pi_{\widehat{\theta}}^\eta}\left[R(\theta_*, x, a) - \frac{1}{\eta}\ln\frac{\pi_{\widehat{\theta}}^\eta(a|x)}{\pi_0(a|x)}\right]$$

$$= \frac{1}{\eta}\mathbb{E}_{\pi_{\theta_*}^\eta}\left[\ln\frac{\pi_0(a|x)\cdot\exp\big(\eta R(\theta_*, x, a)\big)}{\pi_{\theta_*}^\eta(a|x)}\right] - \frac{1}{\eta}\mathbb{E}_{\pi_{\widehat{\theta}}^\eta}\left[\ln\frac{\pi_0(a|x)\cdot\exp\big(\eta R(\theta_*, x, a)\big)}{\pi_{\widehat{\theta}}^\eta(a|x)}\right]$$

$$= \frac{1}{\eta}\mathbb{E}_{x\sim d_0}\big[\ln Z_{\theta_*}^\eta(x)\big] - \frac{1}{\eta}\mathbb{E}_{x\sim d_0}\big[\ln Z_{\widehat{\theta}}^\eta(x)\big] - \mathbb{E}_{x\sim d_0}\left[\sum_{a\in\mathcal{A}}\pi_{\widehat{\theta}}^\eta(a|x)\cdot\big(R(\theta_*, x, a) - R(\widehat{\theta}, x, a)\big)\right]$$

For an arbitrary reward function $f : \mathcal{X} \times \mathcal{A} \to \mathbb{R}$, let $\Delta(x, a) = f(x, a) - R(\theta_*, x, a)$. Consider the following first derivative of $J(f) = \ln Z_f^\eta(x) - \eta\sum_{a\in\mathcal{A}}\pi_f^\eta(a|x)\cdot\Delta(x, a)$, where $Z_f^\eta(x) = \sum_{a\in\mathcal{A}}\pi_0(a|x)\cdot\exp(\eta\cdot f(x, a))$ and $\pi_f^\eta(a|x) \propto \pi_0(a|x)\cdot\exp(\eta\cdot f(x, a))$.

$$\frac{\partial}{\partial\Delta(x, a)}\left[\ln Z_f^\eta(x) - \eta\sum_{a\in\mathcal{A}}\pi_f^\eta(a|x)\cdot\Delta(x, a)\right]$$

$$= \frac{1}{Z_f^\eta(x)}\cdot\pi_0(a|x)\exp\big(\eta\cdot f(x, a)\big)\cdot\eta - \eta\cdot\pi_f^\eta(a|x)$$

$$- \eta\cdot\Delta(x, a)\cdot\frac{\pi_0(a|x)\cdot\exp\big(\eta\cdot f(x, a)\big)}{Z_f^\eta(x)}\cdot\eta + \eta\cdot\Delta(x, a)\cdot\frac{\big[\pi_0(a|x)\cdot\exp\big(\eta\cdot f(x, a)\big)\big]^2}{[Z_f^\eta(x)]^2}\cdot\eta$$

$$+ \eta\sum_{a'\in\mathcal{A}\backslash\{a\}}\frac{\pi_0(a'|x)\cdot\exp\big(\eta\cdot f(x, a')\big)}{Z_f^\eta(x)}\cdot\eta\cdot\Delta(x, a')\cdot\frac{\pi_0(a|x)\cdot\exp\big(\eta\cdot f(x, a)\big)}{Z_f^\eta(x)}$$

$$= -\eta^2\pi_f^\eta(a|x)\Delta(x, a) + \eta^2[\pi_f^\eta(a|x)]^2\cdot\Delta(x, a) + \eta^2\sum_{a'\in\mathcal{A}\backslash\{a\}}\pi_f^\eta(a'|x)\pi_f^\eta(a|x)\Delta(x, a').$$

Therefore, there exists $f(\cdot, \cdot) = \gamma R(\widehat{\theta}, \cdot, \cdot) + (1 - \gamma)R(\theta_*, \cdot, \cdot)$ such that $\quad(\gamma \in (0, 1))$

$$\mathbb{E}_{x\sim d_0}\big[J(R(\widehat{\theta}, \cdot, \cdot)) - J(R(\theta_*, \cdot, \cdot))\big] = \frac{1}{\eta}\mathbb{E}_{x\sim d_0}\left[-\eta^2\sum_{a\in\mathcal{A}}\pi_f^\eta(a|x)\cdot\gamma\cdot\big(R(\widehat{\theta}, x, a) - R(\theta_*, x, a)\big)^2\right]$$

$$+ \frac{1}{\eta} \mathbb{E}_{x \sim d_0} \Big[ \gamma \eta^2 \sum_{a_1 \in \mathcal{A}} \sum_{a_2 \in \mathcal{A}} \pi_f^\eta(a_1|x) \pi_f^\eta(a_2|x) \big( R(\widehat{\theta}, x, a_1) - R(\theta_*, x, a_1) \big) \big( R(\widehat{\theta}, x, a_2) - R(\theta_*, x, a_2) \big) \Big]$$

$$\geq -\eta \cdot \mathbb{E}_{\pi_f^\eta} \big[ \big( R(\widehat{\theta}, x, a) - R(\theta_*, x, a) \big)^2 \big]$$

From Lemma C.4, if $m \geq 128 \eta^2 D^2 B^2 \cdot \log(2 N_\mathcal{R}(\epsilon_c)/\delta)$, for any $(x, a) \in \mathcal{X} \times \mathcal{A}$ such that $\pi_0(a|x) > 0$, it holds that

$$\eta |R(\widehat{\theta}_0, x, a) - R(\theta_*, x, a)| \leq 1, \quad \eta |R(\widehat{\theta}, x, a) - R(\theta_*, x, a)| \leq 1,$$

which means that for any $(x, a) \in \mathcal{X} \times \mathcal{A}$

$$\frac{\pi_f^\eta(a|x)}{\pi_{\widehat{\theta}_0}^\eta(a|x)} \leq e^4.$$

Let $\epsilon_c = \min \{ \frac{\epsilon}{(1 + c_{m,n}^{-1})B}, \frac{1}{8(1 + c_{m,n})B\eta^2 D^2}, B \}$. From Lemma C.3, if $m \geq 128 \eta^2 D^2 B^2 \cdot \log(2 N_\mathcal{R}(\epsilon_c)/\delta)$ and $n \geq \eta/\epsilon \cdot B^2 \log(N_\mathcal{R}(\epsilon_c)/\delta))$ and $n = c_{m,n} m$ then with high probability the output policy $\pi_{\widehat{\theta}}^\eta$ is $O(\epsilon)$ optimal. $\qquad \square$

### C.3  PROOF OF COROLLARY 3.5

*Proof of Corollary 3.5.* The proof follows the same lines as Theorem 4.4 by replacing the global-coverage condition with the local-coverage condition. It still holds that

$$Q(\pi^*) - Q(\pi_{\widehat{\theta}_0}^\eta) \leq \eta \cdot \mathbb{E}_{\pi_f^\eta} \big[ \big( R(\widehat{\theta}_0, x, a) - R(\theta_*, x, a) \big)^2 \big]$$

where $\pi_f^\eta(a|x) \propto \pi_0(a|x) \cdot \exp(\eta \cdot f(x, a))$ and $f(\cdot, \cdot) = \gamma R(\widehat{\theta}_0, \cdot, \cdot) + (1 - \gamma) R(\theta_*, \cdot, \cdot)$ for some $\gamma \in (0, 1)$. Thus, We have $\mathrm{KL}(\pi_f^\eta(a|x) \| \pi_0) \leq 2\eta B$, which further implies that

$$Q(\pi^*) - Q(\pi_{\widehat{\theta}}^\eta) \leq \eta \cdot C_{\rho_{\mathrm{KL}}} \cdot O \big( \frac{1}{n} B \log(N_\mathcal{R}(\epsilon_c)/\delta) + B(1 + c_{m,n}^{-1}) \epsilon_c \big)$$

by Lemma D.4. Then we can conclude by substituting the value of $m$ into the suboptimality gap. $\quad \square$

## D  PROOFS FROM SECTION 4

### D.1  PROOF OF THEOREM 4.3

*Proof of Theorem 4.3.* The proof follows a similar construction as the one for Theorem 3.1. Consider a simple case when $|\mathcal{X}| = M$ and $|\mathcal{A}| = 2$. We suppose that the context $x$ is drawn uniformly from $\mathcal{X}$ at the beginning of each round. Let $\Theta$ be the set consisting of mappings from $\mathcal{X}$ to $\mathcal{A} = \{0, 1\}$. For each $\theta \in \Theta$, we have $R(\theta, x, a) = \begin{cases} c & \text{if } a = \theta(x), \\ 0 & \text{if } a \neq \theta(x), \end{cases}$ where $c > 0$ is a constant, and $\theta(x)$ is the optimal action under context $x$ when the model is $\theta$.

We pick a pair of model $\theta_1, \theta_2$ in $\Theta$, such that $\theta_1(x) = \begin{cases} \theta_2(x) & \text{if } x \neq x_0, \\ 1 - \theta_2(x) & \text{if } x = x_0. \end{cases}$

We denote by $\mathbb{P}_\theta, \mathbb{E}_\theta$ the probability measure and expectation under the model $\theta$.

Applying Pinsker's inequality (Lemma E.3), we have for all event $A$ measurable with respect to the filtration generated by the observations,

$$|\mathbb{P}_{\theta_1}(A) - \mathbb{P}_{\theta_2}(A)| \leq \sqrt{\log(1/2 + e^c/4 + e^{-c}/4) \mathbb{E}_{\theta_1}[N(x_0)]} \leq \sqrt{c^2 \mathbb{E}_{\theta_1}[N(x_0)]} = \sqrt{c^2 T/M},$$

where the first inequality follows from the chain rule of KL divergence, and the fact that $\mathbb{E}_{\theta_1}[N(x_0)] = T/M$.

Set $A$ to be the event that $\pi_{out}(\theta_1(x_0)|x_0) > 1/2$. Then we have

$$\mathbb{P}_{\theta_1}(\pi_{out}(\theta_1(x_0)|x_0) \leq 1/2) + \mathbb{P}_{\theta_2}(\pi_{out}(\theta_2(x_0)|x_0) \leq 1/2) \geq 1 - |\mathbb{P}_{\theta_1}(A) - \mathbb{P}_{\theta_2}(A)| \geq 1 - \sqrt{c^2 T/M}.$$

If the model $\theta$ is uniformly drawn from $\Theta$, then we have

$$\mathbb{E}_{\theta \sim Unif(\Theta)} \mathbb{P}_\theta (\pi_{out}(\theta(x_0)) \leq 1/2) \geq \frac{1}{2} - \sqrt{c^2 T/4M}$$

for an arbitrary $x_0$.

Then we consider the following suboptimality gap:

$$\mathbb{E}_{\pi_{\theta_*}^\eta} \left[ R(\theta_*, x, a) - \frac{1}{\eta} \ln \frac{\pi_{\theta_*}^\eta(a|x)}{\pi_0(a|x)} \right] - \mathbb{E}_{\pi_{out}} \left[ R(\theta_*, x, a) - \frac{1}{\eta} \ln \frac{\pi_{out}(a|x)}{\pi_0(a|x)} \right]$$

$$= \frac{1}{\eta} \mathbb{E}_{\pi_{\theta_*}^\eta} \left[ \ln \frac{\pi_0(a|x) \cdot \exp\big(\eta R(\theta_*, x, a)\big)}{\pi_{\theta_*}^\eta(a|x)} \right] - \frac{1}{\eta} \mathbb{E}_{\pi_{out}} \left[ \ln \frac{\pi_0(a|x) \cdot \exp\big(\eta R(\theta_*, x, a)\big)}{\pi_{out}(a|x)} \right]$$

$$= \frac{1}{\eta} \mathbb{E}_{\pi_{out}} \left[ \ln \frac{\pi_{out}(a|x)}{\pi^*(a|x)} \right],$$

where the last equality follows from the fact that $\pi_{\theta_*}^\eta \propto \pi_0(a|x) \cdot \exp(\eta R(\theta_*, x, a))$.

To bound the suboptimality gap, we further have

$$\mathbb{E}_{\theta \sim Unif(\Theta)} \mathbb{E}_{\pi_{out}} \left[ \ln \frac{\pi_{out}(a|x)}{\pi^*(a|x)} \right]$$

$$= \mathbb{E}_{\theta \sim Unif(\Theta)} \frac{1}{M} \sum_{x \in \mathcal{X}} \mathbb{E}_{a \sim \pi_{out}(\cdot|x)} \left[ \ln \frac{\pi_{out}(a|x)}{\pi^*(a|x)} \right]$$

$$\geq \mathbb{E}_{\theta \sim Unif(\Theta)} \frac{1}{M} \sum_{x \in \mathcal{X}} \mathbb{P}_\theta(\pi_{out}(\theta(x)) \leq 1/2) \cdot \left[ \frac{1}{2} \ln \frac{1 + \exp(-\eta c)}{2} + \frac{1}{2} \ln \frac{1 + \exp(\eta c)}{2} \right]$$

$$\geq \left( \frac{1}{2} - \sqrt{c^2 T/4M} \right) \left[ \frac{1}{2} \ln \frac{1 + \exp(-\eta c)}{2} + \frac{1}{2} \ln \frac{1 + \exp(\eta c)}{2} \right] \tag{D.1}$$

Note that

$$\frac{\mathrm{d}}{\mathrm{d}u} \left[ \frac{1}{2} \ln \frac{1 + e^{-u}}{2} + \frac{1}{2} \ln \frac{1 + e^u}{2} \right] \bigg|_{u=0} = \frac{1}{2} \left[ \frac{1}{1 + \exp(-u)} - \frac{1}{1 + \exp(u)} \right] \bigg|_{u=0} = 0,$$

$$\frac{\mathrm{d}^2}{\mathrm{d}u^2} \left[ \frac{1}{2} \ln \frac{1 + e^{-u}}{2} + \frac{1}{2} \ln \frac{1 + e^u}{2} \right] = \frac{\exp(u)}{[1 + \exp(u)]^2}.$$

Thus, applying Taylor's expansion on the right-hand side of (D.1), we have

$$\mathbb{E}_{\theta \sim Unif(\Theta)} \mathbb{E}_{\pi_{out}} \left[ \ln \frac{\pi_{out}(a|x)}{\pi^*(a|x)} \right] \geq \frac{1}{2} \cdot \left( \frac{1}{2} - \sqrt{c^2 T/4M} \right) \eta^2 c^2 \cdot \frac{1}{3 + \exp(\eta c)}$$

When $\epsilon < 1/64\eta$, we can set $c = 8\sqrt{\epsilon/\eta}$. To achieve a suboptimality gap of $\epsilon$, we need to satisfy:

$$\frac{1}{2} \cdot \left( \frac{1}{2} - \sqrt{c^2 T/4M} \right) \eta^2 c^2 \cdot \frac{1}{3 + \exp(\eta c)} \leq \eta \epsilon,$$

indicating that $T \geq \frac{\eta M}{512\epsilon} = \Omega(\frac{\eta M}{\epsilon})$.

When $\epsilon \geq 1/64\eta$, or equivalently, $\eta \geq 1/64\epsilon$, we employ a different lower bound for (C.1) as follows:

$$\frac{1}{2} \ln \frac{1 + \exp(-\eta c)}{2} + \frac{1}{2} \ln \frac{1 + \exp(\eta c)}{2} = \frac{1}{2} \ln \frac{2 + \exp(\eta c) + \exp(-\eta c)}{4}$$

$$\geq \frac{1}{2} \cdot \frac{1}{2} \left( \ln \frac{\exp(\eta c) + \exp(-\eta c)}{2} \right)$$

$$\geq \frac{1}{4} (\eta c - \ln 2), \tag{D.2}$$

where the first inequality follows from Jensen's inequality.

Substituting (D.2) into (D.1), we have

$$\epsilon \geq \frac{1}{\eta}\mathbb{E}_{\theta \sim Unif(\Theta)}\mathbb{E}_{\pi_{out}}\left[\ln\frac{\pi_{out}(a|x)}{\pi^*(a|x)}\right] \geq \frac{1}{4}\cdot\left(\frac{1}{2}-\sqrt{c^2T/4M}\right)(\eta c - \ln 2)\cdot\frac{1}{\eta}.$$

Set $c = 64\epsilon$. Then we have $T = \Omega(M/\epsilon^2)$.

□

### D.2 PROOF OF THEOREM 4.4

First, we provide the following lemma for the connection between the likelihood loss and the reward difference, which is a key step to upper bound the reward difference between $\widehat{\theta}$ and $\theta_*$.

**Lemma D.1.** *For an arbitrary policy $\pi$, and a set of context-action pairs $\{(x_i, a_i^1, a_i^2, y_i)\}_{i=1}^n$ generated i.i.d. from the BT model and $\pi$, we have with probability at least $1 - \delta$, for any $s \leq n$,*

$$\frac{1}{2}\sum_{i=1}^{s}\mathcal{L}(\theta|x_i, a_i^1, a_i^2, y_i) - \mathcal{L}(\theta_*|x_i, a_i^1, a_i^2, y_i)$$

$$\leq \log(1/\delta) - \frac{1}{8}e^{-B}\sum_{i=1}^{s}\left([R(\theta, x_i, a_i^2) - R(\theta, x_i, a_i^1)] - [R(\theta_*, x_i, a_i^2) - R(\theta_*, x_i, a_i^1)]\right)^2$$

*Proof.* Applying Lemma E.4 to the sequence

$$\left\{-\frac{1}{2}y_i\cdot\log\frac{\sigma(R(\theta_*, x_i, a_i^1) - R(\theta_*, x_i, a_i^2))}{\sigma(R(\theta, x_i, a_i^1) - R(\theta, x_i, a_i^2))} - \frac{1}{2}(1-y_i)\cdot\log\frac{\sigma(R(\theta_*, x_i, a_i^2) - R(\theta_*, x_i, a_i^1))}{\sigma(R(\theta, x_i, a_i^2) - R(\theta, x_i, a_i^1))}\right\}_{i=1}^{n},$$

We have with probability at least $1 - \delta$, for all $s \leq n$,

$$\frac{1}{2}\sum_{i=1}^{s}\mathcal{L}(\theta|x_i, a_i^1, a_i^2, y_i) - \mathcal{L}(\theta_*|x_i, a_i^1, a_i^2, y_i)$$

$$\leq \log(1/\delta) + \sum_{i=1}^{s}\log\left(\sqrt{\sigma(R(\theta_*, x_i, a_i^2) - R(\theta_*, x_i, a_i^1))\cdot\sigma(R(\theta, x_i, a_i^2) - R(\theta, x_i, a_i^1))}\right.$$

$$\left. + \sqrt{\sigma(R(\theta_*, x_i, a_i^1) - R(\theta_*, x_i, a_i^2))\cdot\sigma(R(\theta, x_i, a_i^1) - R(\theta, x_i, a_i^2))}\right)$$

$$= \log(1/\delta) - \frac{1}{2}\sum_{i=1}^{s}\left(\sqrt{\sigma(R(\theta_*, x_i, a_i^2) - R(\theta_*, x_i, a_i^1))} - \sqrt{\sigma(R(\theta, x_i, a_i^2) - R(\theta, x_i, a_i^1))}\right)^2$$

$$\leq \log(1/\delta) - \frac{1}{8}\sum_{i=1}^{s}\left(\sigma(R(\theta_*, x_i, a_i^2) - R(\theta_*, x_i, a_i^1)) - \sigma(R(\theta, x_i, a_i^2) - R(\theta, x_i, a_i^1))\right)^2$$

$$\leq \log(1/\delta) - \frac{1}{8}e^{-B}\sum_{i=1}^{s}\left([R(\theta, x_i, a_i^2) - R(\theta, x_i, a_i^1)] - [R(\theta_*, x_i, a_i^2) - R(\theta_*, x_i, a_i^1)]\right)^2,$$

where the equality follows from the fact that $\sigma(r) + \sigma(-r) = 1$ and the last inequality holds since $\sigma'(r) = \sigma(r)\cdot(1 - \sigma(r)) \geq e^{-B}$ for all $r \in [-B, B]$. □

To further control the error bound for the reward function with the help of Lemma D.1, we introduce the following lemma to show that the likelihood function class $\mathcal{L}$ can be well-covered by the $\epsilon$-net of the reward function class $\mathcal{R}$.

**Lemma D.2** (Covering number of $\mathcal{L}$). *For any $\epsilon_c > 0$, consider an $\epsilon_c$-net $\mathcal{R}^c = \{R(\theta, \cdot, \cdot)|\theta \in \Theta^c\}$ for the reward function class $\mathcal{R}$ with size $N_{\mathcal{R}}(\epsilon_c)$. Then for any $\theta \in \Theta$, there exists $\theta^c \in \Theta^c$ such that for any $s \in [n]$,*

$$\sum_{i=1}^{s}\mathcal{L}(\theta|x_i, a_i^1, a_i^2, y_i) \leq \sum_{i=1}^{s}\mathcal{L}(\theta^c|x_i, a_i^1, a_i^2, y_i) + 2s\epsilon_c.$$

*Proof.* For any $r \in \mathbb{R}$, we have

$$\frac{\mathrm{d}\log(\sigma(r))}{\mathrm{d}r} = \frac{1}{\sigma(r)} \cdot \sigma(r) \cdot (1 - \sigma(r)) = 1 - \sigma(r) \in (0, 1).$$

Thus, for any $\theta \in \Theta$, $x \in \mathcal{X}$, $a^1, a^2 \in \mathcal{A}$ and $y \in \{0, 1\}$, there exists $\theta^c \in \Theta^c$ such that

$$\begin{aligned}
&\left| \mathcal{L}(\theta | x, a^1, a^2, y) - \mathcal{L}(\theta^c | x, a^1, a^2, y) \right| \\
&\leq \left| [R(\theta, x, a^1) - R(\theta, x, a^2)] - [R(\theta^c, x, a^1) - R(\theta^c, x, a^2)] \right| = 2\epsilon_c.
\end{aligned}$$

$\square$

With the above two lemmas, we are now ready to provide the confidence bound for the MLE estimator of the reward function.

**Lemma D.3.** *Consider a set of context-action pairs $\{(x_i, a_i^1, a_i^2, y_i)\}_{i=1}^n$ where labels $\{y_i\}_{i=1}^n$ are generated independently from the BT model. Suppose that $\widehat{\theta}$ is the MLE estimator as defined in Definition 4.1. We have with probability at least $1 - \delta$,*

$$\sum_{i=1}^n \left( [R(\widehat{\theta}, x_i, a_i^2) - R(\widehat{\theta}, x_i, a_i^1)] - [R(\theta_*, x_i, a_i^2) - R(\theta_*, x_i, a_i^1)] \right)^2 \leq O(e^B \log(N_{\mathcal{R}}(\epsilon_c)/\delta) + e^B n\epsilon_c).$$

*Proof.* By Lemma D.1 and Lemma D.2, we have with probability at least $1 - \delta$, for any $\theta \in \Theta$,

$$\frac{1}{2}\sum_{i=1}^n \mathcal{L}(\theta | x_i, a_i^1, a_i^2, y_i) - \mathcal{L}(\theta_* | x_i, a_i^1, a_i^2, y_i)$$

$$\leq \log(N_{\mathcal{R}}(\epsilon_c)/\delta) - \frac{1}{8}e^{-B}\sum_{i=1}^n \left( [R(\theta, x_i, a_i^2) - R(\theta, x_i, a_i^1)] - [R(\theta_*, x_i, a_i^2) - R(\theta_*, x_i, a_i^1)] \right)^2 + O(n\epsilon_c).$$

Since $\widehat{\theta}$ is the MLE estimator, we have $\sum_{i=1}^n \mathcal{L}(\theta | x_i, a_i^1, a_i^2, y_i) - \mathcal{L}(\theta_* | x_i, a_i^1, a_i^2, y_i) \geq 0$, which further implies

$$0 \leq \log(N_{\mathcal{R}}(\epsilon_c)/\delta) - \frac{1}{8}e^{-B}\sum_{i=1}^n \left( [R(\theta, x_i, a_i^2) - R(\theta, x_i, a_i^1)] - [R(\theta_*, x_i, a_i^2) - R(\theta_*, x_i, a_i^1)] \right)^2 + O(n\epsilon_c).$$

Then we have

$$\sum_{i=1}^n \left( [R(\widehat{\theta}, x_i, a_i^2) - R(\widehat{\theta}, x_i, a_i^1)] - [R(\theta_*, x_i, a_i^2) - R(\theta_*, x_i, a_i^1)] \right)^2 \leq O(e^B \log(N_{\mathcal{R}}(\epsilon_c)/\delta) + e^B n\epsilon_c).$$

$\square$

Finally, we provide the on-policy confidence bound for the squared reward difference between the MLE estimator $\widehat{\theta}$ and the optimal reward function $\theta_*$.

**Lemma D.4.** *Consider an arbitrary policy $\pi$, and a set of context-action pairs $\{(x_i, a_i^1, a_i^2, y_i)\}_{i=1}^n$ generated i.i.d. from the BT model and $\pi$. Suppose that $\widehat{\theta}$ is the MLE estimator. We have with probability at least $1 - 2\delta$, there exists a mapping $b : \mathcal{X} \to \mathbb{R}$ such that*

$$\mathbb{E}_\pi \left[ \left( R(\widehat{\theta}, x, a) - R(\theta_*, x, a) - b(x) \right)^2 \right] \leq O\left( \frac{1}{n}e^B \log(N_{\mathcal{R}}(\epsilon_c)/\delta) + e^B \epsilon_c \right).$$

*Proof.* By Lemma D.3, we have with probability at least $1 - \delta$,

$$\sum_{i=1}^n \left( [R(\widehat{\theta}, x_i, a_i^2) - R(\widehat{\theta}, x_i, a_i^1)] - [R(\theta_*, x_i, a_i^2) - R(\theta_*, x_i, a_i^1)] \right)^2 \leq O(e^B \log(N_{\mathcal{R}}(\epsilon_c)/\delta) + e^B n\epsilon_c).$$

We consider an $\epsilon_c$-net $\mathcal{R}^c = \{R(\theta, \cdot, \cdot) | \theta \in \Theta^c\}$ for the reward function class $\mathcal{R}$ with size $N_{\mathcal{R}}(\epsilon_c)$. For any $R(\theta, \cdot, \cdot)$, there exists $R(\theta^c, \cdot, \cdot)$ such that

$$\left| R(\theta, x, a) - R(\theta^c, x, a) \right| \leq O(\epsilon_c)$$

for all $x \in \mathcal{X}, a \in \mathcal{A}$.

Applying Lemma E.1, with probability at least $1 - \delta$, we have

$$
\sum_{i=1}^{n} \left( [R(\theta^c, x_i, a_i^2) - R(\theta^c, x_i, a_i^1)] - [R(\theta_*, x_i, a_i^2) - R(\theta_*, x_i, a_i^1)] \right)^2
$$

$$
- n \mathbb{E}_{x \sim d_0} \mathbb{E}_{a^1, a^2 \sim \pi} \left[ \left( R(\theta^c, x, a^1) - R(\theta_*, x, a^1) - R(\theta^c, x, a^2) + R(\theta_*, x, a^2) \right)^2 \right]
$$

$$
\leq \sqrt{\sum_{i=1}^{n} 4B^2 \mathbb{E}_{x \sim d_0} \mathbb{E}_{a^1, a^2 \sim \pi} \left[ \left( R(\theta^c, x, a^1) - R(\theta_*, x, a^1) - R(\theta^c, x, a^2) + R(\theta_*, x, a^2) \right)^2 \right] \log(N_{\mathcal{R}}(\epsilon_c)/\delta)}
$$

$$
+ \frac{8}{3} B^2 \log(N_{\mathcal{R}}(\epsilon_c)/\delta)
$$

for all $\theta^c \in \Theta^c$.

From Lemma E.2 and the definition of $\Theta^c$, we further have

$$
\mathbb{E}_{x \sim d_0} \mathbb{E}_{a^1, a^2 \sim \pi} \left[ \left( R(\widehat{\theta}, x, a^1) - R(\theta_*, x, a^1) - R(\widehat{\theta}, x, a^2) + R(\theta_*, x, a^2) \right)^2 \right]
$$

$$
\leq O\left(\frac{1}{n} B^2 \log(N_{\mathcal{R}}(\epsilon_c)/\delta) + \frac{1}{n} \sum_{i=1}^{n} \left( [R(\widehat{\theta}, x_i, a_i^2) - R(\widehat{\theta}, x_i, a_i^1)] - [R(\theta_*, x_i, a_i^2) - R(\theta_*, x_i, a_i^1)] \right)^2 + B\epsilon_c \right),
$$

$$
\tag{D.3}
$$

from which we can further derive that

$$
\mathbb{E}_{x \sim d_0} \mathbb{E}_{a^1, a^2 \sim \pi} \left[ \left( R(\widehat{\theta}, x, a^1) - R(\theta_*, x, a^1) - R(\widehat{\theta}, x, a^2) + R(\theta_*, x, a^2) \right)^2 \right]
$$

$$
\leq O\left( \frac{1}{n} e^B \log(N_{\mathcal{R}}(\epsilon_c)/\delta) + e^B \epsilon_c \right)
$$

with probability at least $1 - 2\delta$ from Lemma D.3 and the union bound.

We can then complete the proof by setting

$$
b(x) = \mathbb{E}_{a^2 \sim \pi(\cdot|x)} \left[ R(\widehat{\theta}, x, a^2) - R(\theta_*, x, a^2) \right].
$$

$\square$

**Lemma D.5** (Coverage of $\pi_*$ and $\pi_{\widehat{\theta}}$ by $\pi_{\widehat{\theta}_0}$). *If $m \geq 32\eta^2 D^2 e^B \log(N_{\mathcal{R}}(\epsilon_c))$, $n = c_{m,n} m$ and $\epsilon_c \leq \frac{1}{(1 + c_{m,n}) e^B \eta^2 D^2}$ in Algorithm 2 and Assumption 2.6 holds, then with probability at least $1 - 4\delta$, there exists $b_1 : \mathcal{X} \to \mathbb{R}$ and $b_2 : \mathcal{X} \to \mathbb{R}$ such that*

$$
\eta |R(\widehat{\theta}_0, x, a) - R(\theta_*, x, a) - b_1(x)| \leq 1, \quad \eta |R(\widehat{\theta}, x, a) - R(\theta_*, x, a) - b_2(x)| \leq 1
$$

*for all $x \in \mathcal{X}, a \in \mathcal{A}$ such that $\pi_0(a|x) > 0$.*

*Proof.* By Lemma D.3 and the union bound, we have with probability at least $1 - \delta$, it holds that

$$
\sum_{i=1}^{m} \left( [R(\widehat{\theta}, \widetilde{x}_i, \widetilde{a}_i^2) - R(\widehat{\theta}, \widetilde{x}_i, \widetilde{a}_i^1)] - [R(\theta_*, \widetilde{x}_i, \widetilde{a}_i^2) - R(\theta_*, \widetilde{x}_i, \widetilde{a}_i^1)] \right)^2
$$

$$
+ \sum_{i=1}^{n} \left( [R(\widehat{\theta}, x_i, a_i^2) - R(\widehat{\theta}, x_i, a_i^1)] - [R(\theta_*, x_i, a_i^2) - R(\theta_*, x_i, a_i^1)] \right)^2
$$

$$
\leq O(e^B \log(N_{\mathcal{R}}(\epsilon_c)/\delta) + e^B (n + m)\epsilon_c).
\tag{D.4}
$$

Consider an $\epsilon_c$-net $\mathcal{R}^c = \{R(\theta, \cdot, \cdot) | \theta \in \Theta^c\}$ for the reward function class $\mathcal{R}$ with size $N_{\mathcal{R}}(\epsilon_c)$. For any $R(\theta, \cdot, \cdot)$, there exists $R(\theta^c, \cdot, \cdot)$ such that

$$
\left| R(\theta, x, a) - R(\theta^c, x, a) \right| \leq O(\epsilon_c)
$$

for all $x \in \mathcal{X}, a \in \mathcal{A}$.

Applying Lemma E.1, with probability at least $1 - \delta$, we have

$$\sum_{i=1}^{m} \left( [R(\theta^c, \widetilde{x}_i, \widetilde{a}_i^2) - R(\theta^c, \widetilde{x}_i, \widetilde{a}_i^1)] - [R(\theta_*, x_i, a_i^2) - R(\theta_*, x_i, a_i^1)] \right)^2$$

$$- m\mathbb{E}_{x \sim d_0}\mathbb{E}_{a^1, a^2 \sim \pi_0} \left[ \left( R(\theta^c, x, a^1) - R(\theta_*, x, a^1) - R(\theta^c, x, a^2) + R(\theta_*, x, a^2) \right)^2 \right]$$

$$\leq \sqrt{\sum_{i=1}^{m} 4B^2 \mathbb{E}_{x \sim d_0}\mathbb{E}_{a^1, a^2 \sim \pi_0} \left[ \left( R(\theta^c, x, a^1) - R(\theta_*, x, a^1) - R(\theta^c, x, a^2) + R(\theta_*, x, a^2) \right)^2 \right] \log(N_{\mathcal{R}}(\epsilon_c)/\delta)}$$

$$+ \frac{8}{3}B^2 \log(N_{\mathcal{R}}(\epsilon_c)/\delta)$$

for all $\theta^c \in \Theta^c$.

From Lemma E.2 and the definition of $\Theta^c$, we further have

$$\mathbb{E}_{x \sim d_0}\mathbb{E}_{a^1, a^2 \sim \pi} \left[ \left( R(\widehat{\theta}, x, a^1) - R(\theta_*, x, a^1) - R(\widehat{\theta}, x, a^2) + R(\theta_*, x, a^2) \right)^2 \right]$$

$$\leq O(\frac{1}{m}B^2 \log(N_{\mathcal{R}}(\epsilon_c)/\delta) + \frac{1}{m}\sum_{i=1}^{n} \left( [R(\widehat{\theta}, \widetilde{x}_i, \widetilde{a}_i^2) - R(\widehat{\theta}, \widetilde{x}_i, \widetilde{a}_i^1)] - [R(\theta_*, \widetilde{x}_i, \widetilde{a}_i^2) - R(\theta_*, \widetilde{x}_i, \widetilde{a}_i^1)] \right)^2 + B\epsilon_c).$$

$$(D.5)$$

Substituting (D.4) into (D.5), we have with probability at least $1 - 2\delta$,

$$\mathbb{E}_{x \sim d_0}\mathbb{E}_{a^1, a^2 \sim \pi_0} \left[ \left( R(\widehat{\theta}, x, a^1) - R(\theta_*, x, a^1) - R(\widehat{\theta}, x, a^2) + R(\theta_*, x, a^2) \right)^2 \right]$$

$$\leq O\left( \frac{1}{m}e^B \log(N_{\mathcal{R}}(\epsilon_c)/\delta) + e^B \cdot \frac{n+m}{m} \cdot \epsilon_c \right).$$

Therefore, there exists a mapping $b_2 : \mathcal{X} \to \mathbb{R}$ such that

$$\mathbb{E}_{\pi_0}\left[ \left( R(\widehat{\theta}, x, a) - R(\theta_*, x, a) - b_2(x) \right)^2 \right] \leq O\left( \frac{1}{m}e^B \log(N_{\mathcal{R}}(\epsilon_c)/\delta) + e^B \cdot \frac{n+m}{m} \cdot \epsilon_c \right).$$

From Lemma D.4, we have with probability at least $1 - 2\delta$, there exists a mapping $b_1 : \mathcal{X} \to \mathbb{R}$ such that

$$\mathbb{E}_{\pi_0}\left[ \left( R(\widehat{\theta}_0, x, a) - R(\theta_*, x, a) - b_1(x) \right)^2 \right] \leq O\left( \frac{1}{m}e^B \log(N_{\mathcal{R}}(\epsilon_c)/\delta) + e^B(1 + c_{m,n})\epsilon_c \right).$$

Hence, we can complete the proof by a union bound over the two events and Assumption 2.6. $\qquad\square$

*Proof of Theorem 4.4.* Let $b$ be the mapping defined in Lemma D.4 for $\widehat{\theta}$ We have

$$\mathbb{E}_{\pi_{\theta_*}^{\eta}}\left[ R(\theta_*, x, a) - \frac{1}{\eta}\ln\frac{\pi_{\theta_*}^{\eta}(a|x)}{\pi_0(a|x)} \right] - \mathbb{E}_{\pi_{\widehat{\theta}}^{\eta}}\left[ R(\theta_*, x, a) - \frac{1}{\eta}\ln\frac{\pi_{\widehat{\theta}}^{\eta}(a|x)}{\pi_0(a|x)} \right]$$

$$= \frac{1}{\eta}\mathbb{E}_{\pi_{\theta_*}^{\eta}}\left[ \ln\frac{\pi_0(a|x) \cdot \exp\left(\eta R(\theta_*, x, a)\right)}{\pi_{\theta_*}^{\eta}(a|x)} \right] - \frac{1}{\eta}\mathbb{E}_{\pi_{\widehat{\theta}}^{\eta}}\left[ \ln\frac{\pi_0(a|x) \cdot \exp\left(\eta R(\theta_*, x, a)\right)}{\pi_{\widehat{\theta}}^{\eta}(a|x)} \right]$$

$$= \frac{1}{\eta}\mathbb{E}_{x \sim d_0}\left[ \ln Z_{\theta_*}^{\eta}(x) \right] - \frac{1}{\eta}\mathbb{E}_{x \sim d_0}\left[ \ln Z_{\widehat{\theta}}^{\eta}(x) \right] - \mathbb{E}_{x \sim d_0}\left[ \sum_{a \in \mathcal{A}} \pi_{\widehat{\theta}}^{\eta}(a|x) \cdot \left( R(\theta_*, x, a) - R(\widehat{\theta}, x, a) \right) \right].$$

For an arbitrary reward function $f : \mathcal{X} \times \mathcal{A} \to \mathbb{R}$, let $\Delta(x, a) = f(x, a) - R(\theta_*, x, a)$. Consider the following first derivative of $J(f) = \ln Z_f^{\eta}(x) - \eta \sum_{a \in \mathcal{A}} \pi_f^{\eta}(a|x) \cdot \Delta(x, a)$, where $Z_f^{\eta}(x) = \sum_{a \in \mathcal{A}} \pi_0(a|x) \cdot \exp(\eta \cdot f(x, a))$ and $\pi_f^{\eta}(a|x) \propto \pi_0(a|x) \cdot \exp(\eta \cdot f(x, a))$.

Similar to the proof of Theorem 3.3, we still have

$$\frac{\partial}{\partial\Delta(x, a)}\left[ \ln Z_f^{\eta}(x) - \eta\sum_{a \in \mathcal{A}} \pi_f^{\eta}(a|x) \cdot \Delta(x, a) \right]$$

$$= \frac{1}{Z_f^\eta(x)} \cdot \pi_0(a|x) \exp\big(\eta \cdot f(x,a)\big) \cdot \eta - \eta \cdot \pi_f^\eta(a|x)$$

$$- \eta \cdot \Delta(x,a) \cdot \frac{\pi_0(a|x) \cdot \exp\big(\eta \cdot f(x,a)\big)}{Z_f^\eta(x)} \cdot \eta + \eta \cdot \Delta(x,a) \cdot \frac{\big[\pi_0(a|x) \cdot \exp\big(\eta \cdot f(x,a)\big)\big]^2}{[Z_f^\eta(x)]^2} \cdot \eta$$

$$+ \eta \sum_{a' \in \mathcal{A} \setminus \{a\}} \frac{\pi_0(a'|x) \cdot \exp\big(\eta \cdot f(x,a')\big)}{Z_f^\eta(x)} \cdot \eta \cdot \Delta(x,a') \cdot \frac{\pi_0(a|x) \cdot \exp\big(\eta \cdot f(x,a)\big)}{Z_f^\eta(x)}$$

$$= -\eta^2 \pi_f^\eta(a|x)\Delta(x,a) + \eta^2 [\pi_f^\eta(a|x)]^2 \cdot \Delta(x,a) + \eta^2 \sum_{a' \in \mathcal{A} \setminus \{a\}} \pi_f^\eta(a'|x)\pi_f^\eta(a|x)\Delta(x,a').$$

Note that
$$J(R(\widehat{\theta}, x, \cdot)) = \ln Z_{\widehat{\theta}}^\eta(x) - \eta \sum_{a \in \mathcal{A}} \pi_{\widehat{\theta}}^\eta(a|x) \cdot \big(R(\widehat{\theta}, x, a) - R(\theta_*, x, a)\big)$$

$$= \ln \sum_{a \in \mathcal{A}} \pi_0(a|x) \cdot \exp(\eta(R(\widehat{\theta}, x, a) - b(x))) - \eta \sum_{a \in \mathcal{A}} \pi_{\widehat{\theta}}^\eta(a|x) \cdot \big(R(\widehat{\theta}, x, a) - R(\theta_*, x, a) - b(x)\big)$$

$$= J(R(\widehat{\theta}, x, \cdot) - b(x)).$$

Therefore, there exists $f(\cdot, \cdot) = \gamma[R(\widehat{\theta}, \cdot, \cdot) - b(\cdot)] + (1 - \gamma)R(\theta_*, \cdot, \cdot)$ such that $\quad (\gamma \in (0,1))$

$$\mathbb{E}_{x \sim d_0}[J(R(\widehat{\theta}, \cdot, \cdot)) - J(R(\theta_*, \cdot, \cdot))]$$

$$= \frac{1}{\eta}\mathbb{E}_{x \sim d_0}\bigg[-\eta^2 \sum_{a \in \mathcal{A}} \pi_f^\eta(a|x) \cdot \gamma \cdot \big(R(\widehat{\theta}, x, a) - R(\theta_*, x, a) - b(x)\big)^2\bigg]$$

$$+ \frac{1}{\eta}\mathbb{E}_{x \sim d_0}\bigg[\gamma\eta^2 \sum_{a_1 \in \mathcal{A}} \sum_{a_2 \in \mathcal{A}} \pi_f^\eta(a_1|x)\pi_f^\eta(a_2|x)\big(R(\widehat{\theta}, x, a_1) - R(\theta_*, x, a_1) - b(x)\big)$$

$$\big(R(\widehat{\theta}, x, a_2) - R(\theta_*, x, a_2) - b(x)\big)\bigg]$$

$$\geq -\eta \cdot \mathbb{E}_{\pi_f^\eta}\big[\big(R(\widehat{\theta}, x, a) - R(\theta_*, x, a) - b(x)\big)^2\big]$$

From Lemma D.2, if $m \geq 32\eta^2 D^2 e^B \cdot \log(2N_{\mathcal{R}}(\epsilon_c)/\delta)$, for any $(x,a) \in \mathcal{X} \times \mathcal{A}$ such that $\pi_0(a|x) > 0$, it holds that
$$\eta|R(\widehat{\theta}_0, x, a) - R(\theta_*, x, a) - b_1(x)| \leq 1, \quad \eta|R(\widehat{\theta}, x, a) - R(\theta_*, x, a) - b_2(x)| \leq 1,$$
which means that
$$\frac{\pi_f^\eta}{\pi_{\widehat{\theta}_0}^\eta} \leq e^4.$$

Let $\epsilon_c = \min\{\frac{\epsilon}{2(1+c_{m,n}^{-1})e^B}, \frac{1}{(1+c_{m,n})e^B\eta^2 D^2}\}$. From Lemma D.4, under the condition of the theorem, with high probability the output policy $\pi_{\widehat{\theta}}^\eta$ is $O(\epsilon)$ optimal. $\qquad \square$

### D.3 PROOF OF COROLLARY 4.7

In this subsection, we also discuss our result under the local-coverage condition (Definition 2.8).

*Proof of Corollary 4.7.* The proof follows the same lines as Theorem 4.4 by replacing the global-coverage condition with the local-coverage condition. It still holds that
$$Q(\pi^*) - Q(\pi_{\widehat{\theta}_0}^\eta) \leq \eta \cdot \mathbb{E}_{\pi_f^\eta}\big[\big(R(\widehat{\theta}_0, x, a) - R(\theta_*, x, a) - b(x)\big)^2\big]$$
where $\pi_f^\eta(a|x) \propto \pi_0(a|x) \cdot \exp(\eta \cdot f(x,a))$ and $f(\cdot, \cdot) = \gamma[R(\widehat{\theta}_0, \cdot, \cdot) - b(\cdot)] + (1 - \gamma)R(\theta_*, \cdot, \cdot)$ for some $\gamma \in (0,1)$. Thus, We have $\mathrm{KL}(\pi_f^\eta(a|x)\|\pi_0) \leq 2\eta B$, which further implies that
$$Q(\pi^*) - Q(\pi_{\widehat{\theta}}^\eta) \leq \eta \cdot C_{\rho_{\mathrm{KL}}} \cdot O\big(\frac{1}{n}e^B \log(N_{\mathcal{R}}(\epsilon_c)/\delta) + e^B(1 + c_{m,n}^{-1})\epsilon_c\big)$$
by Lemma D.4. Then we can conclude by substituting the value of $m$ into the suboptimality gap. $\quad \square$

# E   AUXILIARY LEMMAS

**Lemma E.1** (Freedman inequality). *Let $M, v > 0$ be fixed constants. Let $\{X_i\}_{i=1}^n$ be a stochastic process, $\{\mathcal{G}_i\}_i$ be a sequence of $\sigma$-fields, and $X_i$ be $\mathcal{G}_i$-measurable, while almost surely*

$$\mathbb{E}[X_i|\mathcal{G}_i] = 0, |X_i| \leq M, \text{ and } \sum_{i=1}^n \mathbb{E}[X_i^2|\mathcal{G}_{i-1}] \leq v.$$

*Then for any $\delta > 0$, with probability at least $1 - \delta$, it holds that*

$$\sum_{i=1}^n X_i \leq \sqrt{2v \log(1/\delta)} + \frac{2}{3} M \log(1/\delta).$$

**Lemma E.2.** *Suppose $a, b \geq 0$. If $x^2 \leq a + b \cdot x$, then $x^2 \leq 2b^2 + 2a$.*

*Proof.* By solving the root of quadratic polynomial $q(x) := x^2 - b \cdot x - a$, we obtain $\max\{x_1, x_2\} = (b + \sqrt{b^2 + 4a})/2$. Hence, we have $x \leq (b + \sqrt{b^2 + 4a})/2$ provided that $q(x) \leq 0$. Then we further have

$$x^2 \leq \frac{1}{4}\left(b + \sqrt{b^2 + 4a}\right)^2 \leq \frac{1}{4} \cdot 2\left(b^2 + b^2 + 4a\right) \leq 2b^2 + 2a. \tag{E.1}$$

$\square$

**Lemma E.3** (Pinsker's inequality). *If $\mathbb{P}_1, \mathbb{P}_2$ are two probability measures on a common measurable space $(\Omega, \mathcal{F})$, then it holds that*

$$\delta(\mathbb{P}_1, \mathbb{P}_2) \leq \sqrt{\frac{1}{2}\mathrm{KL}(\mathbb{P}_1 \| \mathbb{P}_2)},$$

*where $\delta(\cdot, \cdot)$ is the total variation distance and $\mathrm{KL}(\cdot \| \cdot)$ is the Kullback-Leibler divergence.*

**Lemma E.4** (Lemma A.4, Foster et al. 2021). *For any sequence of real-valued random variables $(X_t)_{t \leq T}$ adapted to a filtration $(\mathcal{F}_t)_{t \leq T}$, it holds that with probability at least $1 - \delta$, for all $T' \leq T$,*

$$\sum_{t=1}^{T'} X_t \leq \sum_{t=1}^{T'} \log\left(\mathbb{E}_{t-1}[e^{X_t}]\right) + \log(1/\delta).$$

