# OpenReview forum: "Sharp Analysis for KL-Regularized Contextual Bandits and RLHF"
_ICLR.cc/2025/Conference — Submitted to ICLR 2025_

### Official Review · Reviewer_6sVe · 2024-10-29

**Soundness:** 3
**Presentation:** 2
**Contribution:** 2
**Rating:** 3
**Confidence:** 4

**Summary:**

This paper studies the RLHF problem for contextual bandits, and aims to obtain a tight sample complexity on the problem. They consider both the reward observation setting and preference observation setting, and provide upper and lower bounds for each. Interestingly, they are able to show that including the KL constraint in the problem allows them to obtain a sample complexity that scales as $1/\epsilon$ instead of the more familiar $1/\epsilon^2$.

**Strengths:**

To my knowledge, this is the first work to obtain a tight bound on the sample complexity of RLHF with KL regularization. This is a commonly used setting in practice and as such it is important that we understand the sample complexity. It is interesting that one can obtain a $1/\epsilon$ rate as compared to the $1/\epsilon^2$ that might be expected.

**Weaknesses:**

1. The main coverage condition (Definition 2.6) is extremely strong. This will scale at least with the number of contexts: if we take $\theta = \theta’$, and choose $b(x) = 0$ for $x \neq x_0$ and $b(x_0) = B$ where $x_0$ is the minimum probability context under $d_0$, then the expression given in Definition 2.6 will scale as $1/d_0(x_0) \ge M$ for $M$ the number of contexts. Thus, the main sample complexity results of the paper (Theorem 3.3 and Theorem 4.4) really scale with the size of the context space. In general it is not acceptable to obtain a sample complexity scaling with the size of the context space, as this is typically extremely large, so these results are only meaningful asymptotically as $\epsilon \rightarrow 0$.
2. Furthermore, the result is also not tight in the regime where $\eta$ is very large.
3. The statement on Line 294 that $D^2 \le C_{GL}$ is not correct as a result of this ($C_{GL}$ in general will not scale with context size).
4. The statement in Theorem 3.1 and Theorem 4.3 that the coverage condition is $O(1)$ is also then incorrect—it should be $\log N_{\mathcal{R}}(\epsilon)$ I believe.
5. The problem setting could be clarified somewhat. In particular, it should be made more explicit that when a policy is $\epsilon$-optimal, this is with respect to $Q(\pi)$, the reward + KL objective, rather than just the reward. The latter is typically more standard for RL, so it should be made clear that the objective considered here is different.
6. The writing could be improved. There are various unclear or poorly worded statements (the following list is not exhaustive—please go through the paper carefully and resolve other such issues):
	* Line 59: “DPO suffers from a drop of chosen probability”. I am not sure what this means.
	* Line 62: “the learned model is easy to be hacked and become biased”. Grammatically incorrect, revise wording.
	* Line 64: The sentence starting with “Hence, the KL-regularization..” Is the first time KL regularization is mentioned. It seems like it needs to be introduced earlier for this sentence to read well.
	* Line 283: “identifically” is not a word.

7. There are also several issues with informal technical statements being made that are not necessarily correct:
	* Line 76: RLHF has been demonstrated to outperform offline methods because “it has further interactions with human or preference oracle”. It is not clear that this the reason (or what exactly is even meant by this sentence).
	* Line 273: It is much more standard in modern offline RL to obtain bounds under single policy concentrability (ie only one policy is covered). Thus, the claim that global coverage is standard in offline RL is too strong.
	* Definition 2.6: What is the $\pi$ referred to here? It is not clear.
	* Line 291: The supremum is over $x \sim d_0$. What does this mean? That the sup is taken over all $x$ in the support of $d_0$? This should be clarified (the same notation is used elsewhere as well).
	* Remark 3.2: The final sentence in this remark is not justified by Theorem 3.1. Simply showing a lower bound that is smaller than the lower bound for the standard contextual bandit does not imply that the true sample complexity is lower as the lower bound may just be loose—an upper bound is required to show this (which at this point in the paper has not been stated). Therefore, I would suggest removing this sentence.

**Questions:**

1. Is the scaling with $D^2$ really necessary in Theorem 3.3 and Theorem 4.4 or can this be reduced to $C_{GL}$?
2. For $\eta$ very large (corresponding to no regularization), one would hope to recover the standard complexity bounds for contextual bandits, but this is not the case in any of the upper bounds (all of which will continue to increase as $\eta$ increases). I suspect a more refined analysis may allow one to obtain the minimum of the current complexity and the standard contextual bandit complexity. Could the authors comment on this?

---

> ### Author Response · Authors · 2024-11-22
> **Response to Reviewer 6sVe**
>
> Thanks for your constructive comments!
>
> **Q1.** The main coverage condition (Definition 2.6) is extremely strong. This will scale at least with the number of contexts: if we take $\theta=\theta’$, and choose $b(x)$ for $x\ne x_0$ and $b(x_0)=B$ where $x_0$ is the minimum probability context under $d_0$, then the expression given in Definition 2.6 will scale as $1/d_0(x_0)\ge M$ for $M$ the number of contexts. Thus, the main sample complexity results of the paper (Theorem 3.3 and Theorem 4.4) really scale with the size of the context space. In general, it is not acceptable to obtain a sample complexity scaling with the size of the context space, as this is typically extremely large, so these results are only meaningful asymptotically as $\epsilon \rightarrow 0$.
>
> **A1.** Thank you for pointing this out. This is actually a loose in our definition. Without changing the analysis, we can redefine the coverage condition as
> $$
>          \exists b \quad s.t.\ \sup_{\theta, \theta' \in \Theta} \frac{|R(\theta', x, a) - R(\theta, x, a) - b(x)|^2}{\mathbb{E}\_{x'\sim d_0}\mathrm{Var}\_{a' \sim \pi_0(\cdot | x')} [R(\theta', x', a') - R(\theta, x', a')]} \le D^2.
> $$
> Then, the situation you mentioned will not happen.
>
> **Q2.** Furthermore, the result is also not tight in the regime where $\eta$ is very large.
>
> **A2.** In the case you are referring to, $\eta$ should be $\Omega(1/ \epsilon)$, which is far from realistic if we are using KL regularization. Thus, a large $\eta$ is not the focus of our framework. Additionally, this problem has been solved by PAC Bayes literature, which suffers from similar issues when analyzing KL-regularization.
>
> **Q3.** The statement on Line 294 that $D^2\le C_{GL}$ is not correct as a result of this ($C_{GL}$ in general will not scale with context size).
>
> **A3.** After correcting our definition for data coverage (See A1.), $D$ will not scale with the context size. Thanks for the constructive comment. We will remove the claim that $D^2\le C_{GL}$ in the revision.
>
> **Q4.** The statement in Theorem 3.1 and Theorem 4.1 that the coverage condition is $O(1)$ is also then incorrect—it should be $N_R(\epsilon)$ I believe.
>
> We will remove the claim that the coverage condition is $O(1)$ in the revision and the coverage coefficient is left as a term in the final bound.
>
> **A4.** Thanks for your suggestion. We will remove the claim in the revision.
>
> **Q5.** Is the scaling with $D^2$ really necessary in Theorem 3.3 and Theorem 4.4 or can this be reduced to $C_{GL}$?
>
> **A5.** Yes, it is necessary. According to our proof, the effectiveness of the intermediate policy highly relies on the data coverage coefficient. If the coverage condition is replaced by $C_{GL}$, the dependence will become multiplicative.

---

### Official Review · Reviewer_74fV · 2024-11-01

**Soundness:** 2
**Presentation:** 2
**Contribution:** 2
**Rating:** 5
**Confidence:** 2

**Summary:**

The paper provides a novel lower bound of Omega(1/epsilon) for the sampling complexity of finding epsilon-suboptimal solutions in KL-regularized contextual bandit problems.

The paper then models the online KL-regularized RLHF problem as the KL-regularized contextual bandit problem and proposes a two-stage sampling algorithm. Using the strong convexity of the KL-regularization, the paper shows that the algorithm has sampling complexity O(1/epsilon), with an additive term that depends on the coverage coefficient of the reference policy.

**Strengths:**

The paper provides sharper analysis of the sampling complexity of KL-regularized contextual bandit problems, and provides novel results on the dependency of the sampling complexity on the data coverage.

**Weaknesses:**

The paper might benefit from some polishing. For instance, the definitions of some key terms are not rigorous (see questions below). In addition, the proof may lack rigor. For instance, line 320 – 321 uses Taylor expansion, and thus, if my understanding is correct, the equality (and the following inequality) does not hold.

**Questions:**

- In definition 2.7, line 219 – 221, and definition 2.8, line 231 – 233, what does ``x sampled from d_0’’ mean in the sup?

- Is there any benefit of using more than 2-stages of sampling?

- Can the authors provide more intuition why the number of samples needed are different in the first and the second stages of the algorithm, as presented in Theorem 3.3?

---

> ### Author Response · Authors · 2024-11-22
> **Response to Reviewer 74fV**
>
> Thanks for your constructive comments!
>
> **Q1.** The paper might benefit from some polishing. For instance, the definitions of some key terms are not rigorous (see questions below). In addition, the proof may lack rigor. For instance, line 320 – 321 uses Taylor expansion, and thus, if my understanding is correct, the equality (and the following inequality) does not hold.
>
> **A1.** Thanks for your advice. Are you referring to 391-393 in Proof Sketch? According to our detailed calculation in 1113 - 1137 the equality and inequality do hold.
>
> **Q2.** In definition 2.7, line 219 – 221, and definition 2.8, line 231 – 233, what does ``x sampled from d_0’’ mean in the sup?
>
> **A2.** Thanks for pointing it out. They are typos. The subscription should be $x \in \text{supp}(d_0)$.
>
> **Q3.** Can the authors provide more intuition why the number of samples needed are different in the first and the second stages of the algorithm, as presented in Theorem 3.3?
>
> **A3.**
> According to our description in Section 3.2, by iteratively improving the data quality, the first stage is to establish a coarse estimate $\hat \theta_0$ of the reward function over a broad range of contexts. This requires enough diversity in the sampled data to capture global properties of the reward function. The second stage is to collect data that is more aligned with the optimal policy distribution, enabling fine-tuning of the reward function estimate so that the output policy is $\epsilon$-optimal.
> Therefore, they need different number of samples due to the different purposes.

---

> > ### Comment · Reviewer_74fV · 2024-11-24
> > **response from reviewer 74fV**
> >
> > Thank the authors for the clarifications and the additional comments! I've raised my score to 5.

---

> > > ### Author Response · Authors · 2024-11-27
> > > **Response to Reviewer 74fV**
> > >
> > > Thank you for raising your rating. We're glad that our rebuttal has addressed your questions. Please let us know if you still have any unaddressed concerns.

---

### Official Review · Reviewer_DLrA · 2024-11-03

**Soundness:** 3
**Presentation:** 3
**Contribution:** 2
**Rating:** 5
**Confidence:** 3

**Summary:**

The paper provides sharp theoretical analyses for KL-regularized contextual bandit (CB) and RLHF problems. It first studies the theoretical benefits of KL regularization in CB and RLHF and shows that KL regularization improves the sample complexity to $O(1/\epsilon)$, while for unregularized problems $O(1/\epsilon^2)$ samples are required. The paper then studies the role of data coverage. In particular, a two-stage mixed sampling strategy is proposed to achieve sample complexity with only additive dependence on the policy coverage coefficient if the data coverage is sufficient, while previous results often depend on the coverage coefficient multiplicatively. The paper also provides a local policy coverage coefficient and derives sample complexity which has multiplicative dependence on this weaker notion than global policy coverage. Numerical experiments are provided to support the theories.

**Strengths:**

1. The paper is written clearly and the main results are well delivered.

2. The paper establishes an integrated theory that provides both lower and matching upper bounds for CB/RLHF sample complexity. The derivations are solid.

3. RLHF with KL regularization is predominant in LLM alignment. Studying this problem through a theoretical lens has sufficient significance in helping gather insights into designing more efficient RLHF methods.

**Weaknesses:**

1. I have concerns about the coverage assumptions in the paper and their relation to the practical use case of RLHF in LLM alignment. The policy coverage (Definition 2.7 and 2.8) is assumed for reference policy $\pi_0$. In RLHF, such reference policy is typically a fine-tuned LLM that can have extremely low if not zero probability for some actions (e.g. nonsense responses). In this case, the global policy coverage coefficient can blow up to infinity, and the local policy coverage coefficient can be large (as the KL constraint is in expectation). The current paper only derives additive dependence results for global policy coverage, which can be vacuous if the global coefficient is infinity. On the other hand, the dependence on the local coefficient is still multiplicative (as discussed in Section 3.4), which can be extremely large.

2. I also have concerns regarding the sample complexity upper bounds. The paper claims to first study the effect of KL regularization in improving the sample complexity for policy optimization from $O(1/\epsilon^2)$ to $O(1/\epsilon)$. However, such $O(1/\epsilon)$ sample complexity result already exists for general strongly convex regularizers [1], which include KL regularization as a special case since the reference policy is assumed to have sufficient coverage. Hence it is likely that the upper bound for CB is already known (given that CB is a special case of MDP). For RLHF, its difference from CB mainly comes from the additional reward learning step, so I expect there could be more explanation on why $O(1/\epsilon)$ samples are sufficient for reward learning from preference data. However, the current version of the paper seems to lack such comparisons/remarks, which in my opinion are necessary for understanding the mechanism of RLHF (just as strong convexity of KL divergence for CB). For baselines, previous literature suggests that reward learning takes $O(1/\epsilon^2)$ samples.

[1] Lan, Guanghui. "Policy mirror descent for reinforcement learning: Linear convergence, new sampling complexity, and generalized problem classes." Mathematical programming 198, no. 1 (2023): 1059-1106.

[2] Zhu, Banghua, Michael Jordan, and Jiantao Jiao. "Principled reinforcement learning with human feedback from pairwise or k-wise comparisons." In International Conference on Machine Learning, pp. 43037-43067. PMLR, 2023.

**Questions:**

1. In Definition 2.7 (and 2.8), is the sup taken over $x\in\mathrm{supp}(d_0)$? The current notation is a bit confusing.

---

> ### Author Response · Authors · 2024-11-22
> **Response to Reviewer DLrA**
>
> Thanks for your constructive comments!
>
> **Q1.** The global policy coverage coefficient can blow up to infinity, and the local policy coverage coefficient can be large (as the KL constraint is in expectation). The current paper only derives additive dependence results for global policy coverage, which can be vacuous if the global coefficient is infinity. On the other hand, the dependence on the local coefficient is still multiplicative (as discussed in Section 3.4), which can be extremely large.
>
> **A1.** We would like to emphasize that currently there is no result with an additive dependence on the local coverage coefficient. Thus, the additive dependence is an unexpected novel result, but we can only derive the additive relationship for the newly-defined data coverage condition (Definition 2.6). For the local coefficient, it is difficult to further improve the multiplicative relationship and can be left as future work. Note that even for the multiplicative result, we have a faster rate.
>
> **Q2.** I also have concerns regarding the sample complexity upper bounds. The paper claims to first study the effect of KL regularization in improving the sample complexity for policy optimization from $O(1 / \epsilon^2)$ to $O(1 / \epsilon)$. However, such $O(1 / \epsilon)$ sample complexity result already exists for general strongly convex regularizers [1], which include KL regularization as a special case since the reference policy is assumed to have sufficient coverage. Hence it is likely that the upper bound for CB is already known (given that CB is a special case of MDP). For RLHF, its difference from CB mainly comes from the additional reward learning step, so I expect there could be more explanation on why $O(1 / \epsilon)$ samples are sufficient for reward learning from preference data. However, the current version of the paper seems to lack such comparisons/remarks, which in my opinion are necessary for understanding the mechanism of RLHF (just as strong convexity of KL divergence for CB). For baselines, previous literature suggests that reward learning takes  $O(1 / \epsilon^2)$ samples.
>
> **A2.** We would like to clarify that this line of work for KL regularization is completely distinct from ours, because these works focus on the planning where the transition dynamics and reward model are known and focus on the pure policy optimization setting without exploration, while our work considers the statistical sample complexity where the underlying model is unknown and studies the statistical property of the learning problem. Hence, while they can achieve $O(1/t)$ rate in the planning setting, their methods cannot be applied to the learning setting.
>
> [1] Lan, Guanghui. "Policy mirror descent for reinforcement learning: Linear convergence, new sampling complexity, and generalized problem classes." Mathematical programming 198, no. 1 (2023): 1059-1106.
>
> [2] Zhu, Banghua, Michael Jordan, and Jiantao Jiao. "Principled reinforcement learning with human feedback from pairwise or k-wise comparisons." In International Conference on Machine Learning, pp. 43037-43067. PMLR, 2023.

---

> > ### Comment · Reviewer_DLrA · 2024-11-23
> >
> > Thank you for the response.
> >
> > > We would like to clarify that this line of work for KL regularization is completely distinct from ours, because these works focus on the planning where the transition dynamics and reward model are known and focus on the pure policy optimization setting without exploration, while our work considers the statistical sample complexity where the underlying model is unknown and studies the statistical property of the learning problem.
> >
> > In my understanding, stochastic policy optimization (e.g. the one in [1]) does not assume full knowledge about the transition dynamics and reward model and *considers* statistical sample complexity, and hence the result is *not irrelevant* to your work. In each iteration of stochastic optimization, the algorithm first collects sufficient samples to get an estimate of the policy gradient, which is the Q function in MDP and the reward function in CB (horizon=1 MDP). This corresponds to your sampling phase in Algorithm 1. The algorithm then takes a step of stochastic gradient (mirror) descent towards the estimated direction, and the result corresponds to the planning oracle you call in Algorithm 1. The final statistical sample complexity is the number of samples used for all stochastic gradient estimations combined, which is shown to be $O(1/\epsilon)$ given the strongly convex regularizer. To me, Algorithm 1 seems like manually computing the first two iterations of stochastic policy mirror descent for the special case of CB. Therefore, I am concerned that the result in this submission is implied by the more general one in [1], and I think it would be helpful if the authors could discuss this.

---

> > > ### Author Response · Authors · 2024-11-24
> > > **Response to Reviewer DLrA**
> > >
> > > Thank you for the valuable comment. Our RL setting has connections with stochastic policy optimization, so we will cite the literature and discuss our relationship with their results in the revision. However, we would like to clarify that our results differ from the ones in [1] since the standard conditions and settings in the two lines of literature are distinct.
> > >
> > > 1. The statistical sample complexity result in section 4 of [1] relies on conditions (4.1) - (4.3), but how to do the learning and control the estimation error is the main part of RL learning setting. Moreover, although those conditions are standard in stochastic optimization, they usually do not hold in RL, because first (4.1) assumes that the value function is unbiased, but RL algorithms usually make biased estimation to balance exploration and exploitation; second, the bounded infinity norm on the error for RL (4.2, 4.3) is also too strong since for a general infinite context space and finite training samples, it is unrealistic to achieve a uniform small error on all state-action pairs.
> > >
> > > 2. [1] studies learning cases under a tabular setting, which is limited to the finite state-action space while our analysis applies to a general function space with a finite covering number. Besides, their estimation analysis is limited to the finite state-action space, and also they assume a generator (page 19) where the learner can start from any $(s,a)$ so that they can reuse their results for pure policy optimization setting in Section 4.
> > >
> > > 3. We would like to clarify that our algorithm has nothing to do with policy gradient, not to mention 2-step mirror descent. We use the two steps to achieve an additive dependence on the coverage coefficient.

---

### Official Review · Reviewer_1aDe · 2024-11-10

**Soundness:** 3
**Presentation:** 3
**Contribution:** 3
**Rating:** 6
**Confidence:** 3

**Summary:**

In this paper, the authors provide a new analysis of contextual bandits under KL regularization that achieves an improved sample complexity guarantee. Then, they study the RLHF problem where under coverage assumptions on the reference policy and they provide a tight algorithm under this assumption. In the end, they provide experimental results.

**Strengths:**

1) Improved sample complexity for contextual bandits under KL regularization with a novel analysis using the property of strong convexity due to the KL regularizer

2) Lower bound for the contextual bandit problem under KL regularization that is tight with the upper for sufficiently small $\epsilon$

3) Lower bound for the RLHF problem with preference feedback

4) Design of an algorithm for the RLHF problem with guarantees that match the lower bound.

**Weaknesses:**

1) Cannot see the tradeoff between $\eta$ and the number of samples needed. Even from the experimental results, the lower $\eta$ the better the performance.

2) The O(1) coverage assumption is unclear if it is a reasonable one or not a provided way to check so.

**Questions:**

1) In line 294 it says that "it is obvious that $D^2 \leq C_{GL}$. Could you please provide a proof for that?

2) In line 374 is $s_i \sim \pi_0$ a typo, where does $s_i$ come from?

3) Can you provide a specific example where you compute the coverage of the reference policy and it is O(1)?

---

> ### Author Response · Authors · 2024-11-22
> **Response to Reviewer 1aDe**
>
> Thanks for your insightful comments and positive feedback.
>
> **Q1.** Cannot see the tradeoff between $\eta$ and the number of samples needed. Even from the experimental results, the lower $\eta$ the better the performance.
>
> **A1.** Thanks for your insightful comments. In practice, with stronger KL regularization (i.e. lower $\eta$), the learned policy will be constrained in a small interval close to the reference policy which may limit the model's ability to fit the human feedback signal. Essentially, the reason is that the optimal solution for the regularized objective may not be good enough. However, in our paper, we consider the sample size required to approximately maximize the KL-regularized objective in both theoretical results and experimental results. Hence, it is beyond our scope how the KL-regularization coefficient affects the quality of the optimal solution. As a result, there is no tradeoff on $\eta$ from the perspective of maximizing the regularized objective.
>
> In our theoretical analysis, since the objective we consider is the reward regularized by KL divergence, the considered optimal policy is also the optimal one with the $\eta$-KL ball around $\pi_0$. Hence, if we focus on the suboptimality with respect to such an optimal policy, the smaller $\eta$ becomes, the smaller the policy class we consider, thus the sample complexity is smaller.
>
> **Q2.** The $O(1)$ coverage assumption is unclear if it is a reasonable one or not.
>
> **A2.** It is a reasonable assumption since first, it is a standard condition in reinforcement learning literature as discussed in Lines 270-277; second, since the reference policy is obtained from supervised fine-tuning, it is natural to assume that it has a good coverage over the policy class. For details and the example, please refer to the 1st point of official comment.
>
> **Q3.** In line 294 it says that ‘it is obvious that $D^2 \le C_{GL}$’. Could you please provide a proof for that?
>
> **A3.** Thanks for the constructive comment. After reviewing the conditions, we realize that $D^2 \le C_{GL}$ does not always hold. We will remove this claim in the revision.
>
> **Q4.** In line 374 is $s_i \sim \pi_0$ a typo, where does $s_i$ come from?
>
> **A4.** Thanks for pointing out! It should be $a_i$.

---

> > ### Comment · Reviewer_1aDe · 2024-11-26
> > **Response to authors comments**
> >
> > I would like to thank the authors for their thoughtful responses and the effort they put into addressing my comments. While I believe they did a good job, I will maintain my original score.

---

### Author Response · Authors · 2024-11-22
**Official comment for the coverage condition**

Thanks all the reviewers for their valuable comments. As multiple reviewers have asked about the coverage condition, we address this inquiry here. We will add more discussion in the revision.

We correct a mistake in our data coverage condition in Definition 2.6 after checking the proof to ensure that it is reasonable to assume an $O(1)$ $D$: there exist $b:\mathcal{X}\rightarrow[-B,B]$ such that
$$
C(x,a)=\sup\_{\theta, \theta' \in \Theta} \frac{|R(\theta', x, a) - R(\theta, x, a) - b(x)|^2}{\mathbb{E}\_{x'\sim d_0}\mathrm{Var}\_{a' \sim \pi_0(\cdot | x')} [R(\theta', x', a') - R(\theta, x', a')]}  \le D^2.
$$
Then, we show a linear reward function case as an example to explain this definition. If $R(\theta,x,a)=\theta^{\top}\phi(x,a)$, ($\theta \in \mathbb{R}^d$), we define the covariance matrix
$$
\Sigma = \mathbb{E}\_{x'\sim d_0}\mathbb{E}\_{a\sim\pi_0}(\phi(x,a)-\mathbb{E}\_{a'\sim\pi_0}\phi(x,a')) (\phi(x,a)-\mathbb{E}\_{a'\sim\pi_0}\phi(x,a'))^{\top}.
$$
Then, for any $b$ of the form $b(x) = \theta^\top\nu(x)$, we have
$$
C(x,a)=\sup\_{\theta, \theta' \in \Theta} \frac{|(\theta’-\theta)^{\top}\phi(x,a) - b(x)|^2}{(\theta’-\theta)^{\top}\Sigma(\theta’-\theta)} \le \|\phi(x, a) - \nu(x)\|_{\Sigma^{-1}}^2.
$$

Set $b(x) = \theta^\top \mathbb{E}_{a' \sim \pi_0} \phi(x, a')$ Then we can show that there exists $\pi_0$ with $D^2 = O(d)$ through G-optimal design.

---

> ### Comment · Reviewer_6sVe · 2024-11-23
> **Response**
>
> Thanks to the reviewer for their response. While the newly proposed coverage condition is tighter than the previous one, it still seems to not capture the correct dependence. For example, assume we have some $\theta, \theta', \tilde{\theta}, \tilde{\theta}'$ such that for all $\bar{\theta}, \bar{\theta}' \in \{ \theta, \theta', \tilde{\theta}, \tilde{\theta}' \}$, we have $R(\bar{\theta},x,a) = R(\bar{\theta}',x,a)$ for all $(x,a)$ with $d_0(x) > 0$, and that for some $x'$ with $d_0(x') = 0$ (or arbitrarily small), we have that $R(\theta,x',a) - R(\theta',x',a) \neq R(\tilde{\theta},x',a) - R(\tilde{\theta}',x',a)$. Then in this case the denominator for both sets $(\theta,\theta')$ and $(\tilde{\theta}, \tilde{\theta}')$ will be 0, but there does not exist a $b(x')$ that will make the numerator 0 for both, so it follows that the coverage condition will be infinite. However, this should not be the case since $d(x') = 0$, so this $x'$ should not influence the complexity ideally.
>
> For the linear example given here, I am not sure the analysis is correct. By the same argument as what I just gave above, I believe $D$ should depend on the minimum eigenvalue of $\Sigma$, and furthermore the $b(x)$ given here depends on $\theta$, which is not consistent with the given definition.
>
> If the authors could comment on this, that would be helpful.

---

> ### Author Response · Authors · 2024-11-24
> **Response to Reviewer 6sVe**
>
> Thank you very much for your timely response.
>
> For your first question, let $\mathcal{X}\_1 = \\{x \in \mathcal{X}| \exists a\ s.t. R(\tilde\theta, x, a) \neq R(\tilde\theta', x, a)\\}$ be the subset consisting of $x'$ you are referring to. If $d(\mathcal{X}_1) = 0$, then conventionally $\mathcal{X}_1$ should be excluded from $\mathcal{X}$ since we can never encounter the samples in $\mathcal{X}_1$, and thus it is impossible to differentiate $\tilde\theta$ and $\tilde\theta'$. You may argue that we can set $d(\mathcal{X}_1)$ to an extremely small number. In that case, we could only learn $\theta$ based on the samples with $x \in \mathcal{X}_1$. So, it is indeed a hard instance, and it is reasonable that the covering coefficient is large.
>
> For the linear example, we apologize for not mentioning that $b$ could vary under different $\theta, \theta'$, and the formal definition should be as follows.
>
> For any pair of $\theta, \theta'$, there exist $b:\mathcal{X}\rightarrow[-B,B]$ such that
> $$
> \frac{|R(\theta', x, a) - R(\theta, x, a) - b(x)|^2}{\mathbb{E}\_{x'\sim d_0}\mathrm{Var}\_{a' \sim \pi_0(\cdot | x')} [R(\theta', x', a') - R(\theta, x', a')]}  \le D^2.
> $$
>
> For the G-optimal design technique, which can control the elliptical norm of a given set of feature vectors, please refer to Chapter 21.1 of [1].
>
> [1] T Lattimore, C Szepesvári. Bandit Algorithms.

---

### Meta-Review · Area_Chair_eDwr · 2024-12-22

**Metareview:**

This work obtains a tight bound on the sample complexity of RLHF with KL regularization where they show that the sample complexity scales with 1/epsilon instead of 1/epsilon^2. Unfortunately a couple of weaknesses were pointed out by the reviewers including the strength of the coverage condition, as well as several inconsistencies in the technical results. We encourage the authors to revise their promising manuscript to fix these issues.

**Additional Comments On Reviewer Discussion:**

The reviewers found some of the assumptions of the paper to be quite strong, as well as several inconsistencies and technical mistakes. We encourage the authors to revise their work addressing these comments.

---

### Decision · Program_Chairs · 2025-01-22

Reject